# Yeast Cip1 is activated by environmental stress to inhibit Cdk1–G1 cyclins via Mcm1 and Msn2/4

Ya-Lan Chang[1], Shun-Fu Tseng[1,2], Yu-Ching Huang [1], Zih-Jie Shen [1], Pang-Hung Hsu [3], Meng-Hsun Hsieh[1], Chia-Wei Yang[1], Silvia Tognetti[4], Berta Canal[4], Laia Subirana[4], Chien-Wei Wang[1], Hsiao-Tan Chen[1], Chi-Ying Lin[1], Francesc Posas[4] & Shu-Chun Teng[1]

Upon environmental changes, proliferating cells delay cell cycle to prevent further damage accumulation. Yeast Cip1 is a Cdk1 and Cln2-associated protein. However, the function and regulation of Cip1 are still poorly understood. Here we report that Cip1 expression is co-regulated by the cell-cycle-mediated factor Mcm1 and the stress-mediated factors Msn2/4. Overexpression of Cip1 arrests cell cycle through inhibition of Cdk1–G1 cyclin complexes at G1 stage and the stress-activated protein kinase-dependent Cip1 T65, T69, and T73 phosphorylation may strengthen the Cip1and Cdk1–G1 cyclin interaction. Cip1 accumulation mainly targets Cdk1–Cln3 complex to prevent Whi5 phosphorylation and inhibit early G1 progression. Under osmotic stress, Cip1 expression triggers transient G1 delay which plays a functionally redundant role with another hyperosmolar activated CKI, Sic1. These findings indicate that Cip1 functions similarly to mammalian p21 as a stress-induced CDK inhibitor to decelerate cell cycle through G1 cyclins to cope with environmental stresses.

[1] Department of Microbiology, College of Medicine, National Taiwan University, Taipei 10051, Taiwan. [2] Department and Graduate Institute of Microbiology and Immunology, National Defense Medical Center, Taipei 11490, Taiwan. [3] Department of Bioscience and Biotechnology, National Taiwan Ocean University, Keelung 20224, Taiwan. [4] Cell Signaling Research Group, Departament de Ciències Experimentals i de la Salut, Universitat Pompeu Fabra, Barcelona 08003, Spain. Correspondence and requests for materials should be addressed to S.-C.T. (email: shuchunteng@ntu.edu.tw)

Natural environment changes frequently and wild organisms require proper responses to adapt to the challenge. Under environment-friendly condition, cell cycle proceeds without interruption. However, stress conditions temporarily arrest cells in G1 and further promote proper cellular response to overcome the stress and then re-engage cells into the cell cycle[1–4]. For example, depletion of nutrient[5, 6], osmotic stress[7, 8], and oxidative stress-induced DNA damage all lead to G1 arrest for cells to repair the damage[9]. The G1 checkpoint contains several fine-tuned pathways that monitor cell cycle progression.

In budding yeast, the start transition (START) occurs during late G1 and it is usually considered as a critical initiating point of a new round of cell division[10]. START is a point of no return: post-START cells finish cell cycle irrevocably. Execution of START is mediated by a serine/threonine cyclin-dependent kinase (Cdk1) which physically interacts with one of the G1 cyclins (Cln1, Cln2, or Cln3) to form START-promoting protein kinase complex[11, 12]. Cln3 is the most upstream regulatory G1 cyclin in the transition. The Cdk1–Cln3 complex activates SBF and MBF complexes that drive transcription of START genes[13, 14]. SBF associates with its repressor, Whi5, the human Rb analog, at the target promoters to repress transcription. Cdk–Cln3 phosphorylates Whi5 to release Whi5 from SBF and activate START transcription[15, 16]. Under osmotic stress, Whi5 and the co-regulator Msa1 are phosphorylated by the stress-activated protein kinase (SAPK) Hog1 to inhibit G1 cyclin expression for controlling adequate passage through START[17].

On the other hand, cyclin-dependent kinase inhibitor (CKI) interacts with a Cdk1–cyclin complex to block kinase activity, which plays an inhibitory role. In *Saccharomyces cerevisiae*, two CKIs were previously well studied. Far1 is a pheromone-induced Cdk1–G1 cyclin inhibitor[18]. *FAR1* RNA and protein accumulate in early G1 but decline after passing through START[19]. Pheromone treatment triggers the interaction between Far1 and Cdk1–Cln to arrest cell cycle for mating to occur[20]. On the other hand, Sic1, an inhibitor of Cdk1–Clb complex, blocks the activities of Cdk1–Clb5/Clb6 and Cdk1–Clb1/Clb2, which are required for DNA replication and mitosis, respectively[21]. Cdk1–Cln1/Cln2-mediated phosphorylation of Sic1 triggers ubiquitin-dependent proteolysis which initiates DNA replication[22]. Sic1 prevents cells from premature S-phase initiation, which allows cells to have more time to rescue defects[23].

Mcm1 is a MADS family transcription factor[24] that is highly conserved in all eukaryotes[25, 26]. In budding yeast, Mcm1 is required for pheromone response and also regulates the transcription of multiple cell-cycle genes[27–29]. The Mcm1 binding site, the early cell cycle box (ECB) element, controls many M/G1 specific genes transcription[29]. Deletion of the ECBs from *CLN3* and *SWI4* promoters reduces their expression and causes a G1–S delay[30].

Msn2 and its homologous protein, Msn4, are two key transcription factors that regulate the expression of hundreds of stress response element (STRE)-containing stress response genes[31]. Msn2 and Msn4 are activated to evoke rapid and efficient adaptation responses upon several environmental and metabolic cues, including oxidative stress, heat, osmotic stress, DNA replicative stress, carbon source starvation, and diauxic transition[32–35]. Single deletion of *MSN2* or *MSN4* causes no visible phenotype, while *msn2 msn4* mutants are hypersensitive to general stresses[31].

A new G1 CKI, Ypl014w (Cip1) has recently been identified in *Saccharomyces cerevisiae*[36]. Cip1 associates with Cdk1 and Cln2[36]. However, the exact regulatory mechanism of Cip1 to control cell cycle progression is still unclear. Here we demonstrate that Cip1 expression is co-regulated by the cell cycle-mediated

factor Mcm1 and stress-mediated factors Msn2/4. *CIP1* expression fluctuates with cell cycle and peaks at G1. Over-expressed Cip1 binds to all Cdk1–G1 cyclin complexes and blocks cell cycle at G1. At early G1, Mcm1–mediated accumulation of Cip1 mainly targets to Cdk1–Cln3 complex for preventing the inhibitory phosphorylation of Whi5, inhibiting *CLN1/CLN2* expression, and blocking cell cycle progression. Under hyper-osmotic stress, Hog1 phosphorylates Cip1 which might increase the association between Cip1 and Cdk1–G1 cyclin complexes. Cip1 plays a functionally redundant role with another CKI, Sic1. These findings suggest that multiple mechanisms regulate Cip1 expression and activity to control the cell cycle.

## Results

**Overexpression of Cip1 induces G1 arrest.** A previous study demonstrated that Cip1 interacts with the Cdk1–Cln2 complex[36]. To confirm this and further identify the biological function of Cip1, *CIP1* was overexpressed under the *GAL1* promoter. The budding index analysis showed that overexpression of Cip1 increased unbudded cells from 36 to 55% (Fig. 1a). As previously reported[36], Cip1 overexpression caused cell cycle inhibition when cells were released from G1 arrest (Fig. 1b). These data confirm that Cip1 specifically impedes G1 progression when it is highly expressed.

Far1 is a pheromone-induced CKI[18–20], and Sic1 specifically targets Cdk1–Clb under several stresses[21–23]. To determine whether Cip1-induced G1 arrest was Far1- or Sic1-dependent, *FAR1* or *SIC1* was deleted in cells overexpressing Cip1. Cip1 overexpression in wild-type, *far1*, *sic1*, and *far1sic1* strains exhibited no difference on growth retardation (Fig. 1c), suggesting that Far1 and Sic1 are not required for Cip1-induced G1 arrest.

**Cip1 is a CKI of all G1 cyclins.** G1/S transition is controlled by Cdk1–G1 cyclins to activate downstream genes required for S phase initiation. We hypothesized that Cip1 overexpression may reduce Cdk1–G1 cyclin activity. To verify this, we induced G1 cyclin in the Cip1 overexpressed strain. Simultaneous induction of one of G1 cyclins in the Cip1 overexpressed strain partially rescued the Cip1-dependent slow growth through preventing G1 arrest (Fig. 1d). These results suggest that Cip1-mediated G1 arrest is due to the inhibitory targeting of Cip1 to Cdk1–G1 cyclin complexes and this arrest can be rescued by increasing G1 cyclin expression. This is consistent with a previous study showing that Cip1 inhibits Cdk1–Cln2 activity[36]. To further determine whether Cip1 is a direct inhibitor of all three Cdk1–G1 cyclin complexes, we examined whether Cip1 could inhibit the kinase activity of all three Cdk1–G1 cyclins purified from yeast extracts. The presence of GST-Cip1 resulted in a prominent decrease of in vitro Cdk1–G1 cyclin activities (Fig. 1e). These findings indicate that Cip1 directly inhibits all Cdk1–G1 cyclins to block cell cycle.

**CIP1 is expressed by Mcm1 with peak expression at G1.** A large-scale gene expression profile hinted that *CIP1* expression begins at the M/G1 transition[37]. To identify the precise expression pattern of *CIP1* during the cell cycle, G1-synchronized cells were released, and *CIP1* expression was monitored. The expression pattern of *CIP1* mRNA was similar to a G1 cyclin, *CLN2*, which peaked at G1 periodically during cell cycle progression (Fig. 2a,b). Therefore, *CIP1* is a cell cycle-regulated gene mainly expressed at G1.

Mcm1 is a transcription factor that modulates gene expression at M/G1 transition[27–29]. Examination of the upstream sequences revealed two putative ECB elements located at nt −355 to −340

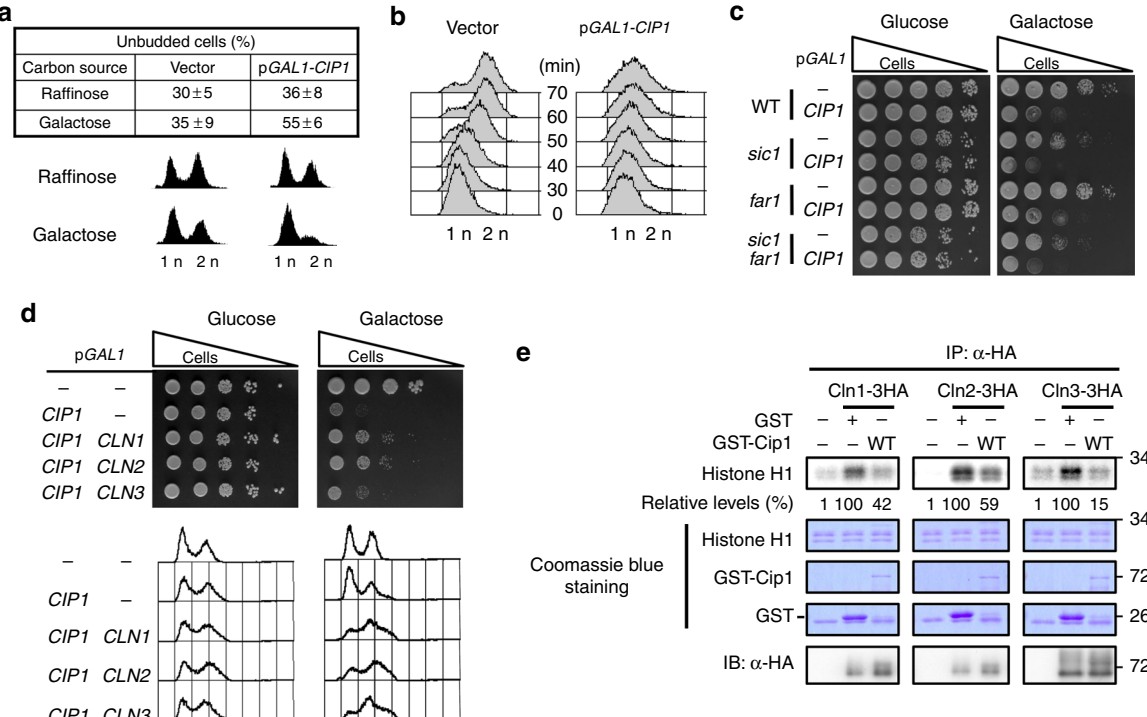

**Fig. 1** Overexpression of *CIP1* causes cell cycle arrest at G1 through inhibition of all Cdk1–G1 cyclin complexes. **a** Strains bearing either the empty vector or *GAL1-CIP1* plasmid were pre-grown in 2% raffinose medium and then added 2% galactose to induce for 9 h. The percentage of unbudded cells was scored. The values were given as mean ± s.d. (*n* = 3). The DNA content was shown below by FACS analysis. **b** Strains with the empty vector or *GAL1-CIP1* plasmids were first arrested at G1 phase by α-factor in 2% raffinose culture. Cip1 was induced in the present of 2% galactose. The α-factor was subsequently removed from the cultures, and samples were collected at the indicated time points for flow cytometry analysis. **c** Isogenic wild-type (WT), *far1*, and *sic1* strains harboring the empty vector or *GAL1-CIP1* plasmid were spotted in 10-fold diluted equal number of yeast on 2% glucose and 2% galactose plates. **d** Strains bearing the *LEU2* empty vector or *GAL1-CIP1* plasmids were co-transformed with the *URA3* empty vector, *GAL1-CLN1*, *GAL1-CLN2*, or *GAL1-CLN3* plasmids. Cells were spotted in 10-fold diluted equal number of yeast on 2% glucose and 2% galactose plates. The FACS analysis of these strains is shown below. **e** The Cdk1-cyclins were immunoprecipitated from strains bearing the empty vector or plasmids with *GAL1* driven 3HA tagged Cln1, Cln2, or Cln3 and incubated with recombinant Cip1. Histone H1 was used for the kinase substrate. The kinase activity was shown as the relative signals of $^{32}$P-labeled histone H1. The immune precipitates were probed with anti-HA antibody to detect the levels of purified cyclins. The uncropped Coomassie *blue images* are shown in Supplementary Fig. 11

and −281 to −266 relative to the initiation codon of Cip1 (Fig. 2c). To determine the contribution of these elements for *CIP1* expression, we generated a chromosomal *lacZ* reporter strain with the 500 bp upstream untranslated region of *CIP1* fused to a *lacZ* gene. The *lacZ* expression was dramatically decreased when both ECB elements were mutated (Fig. 2d), indicating that the ECB elements contribute to *CIP1* transcription. We further generated mutations at the ECB elements of the endogenous *CIP1* promoter. Northern blot analysis showed that the cell cycle-dependent *CIP1* expression was compromised in the *ecb I + II* mutants (Fig. 2a,b). Next, we asked whether *CIP1* is transcriptionally activated through direct Mcm1 binding at ECBs during M/G1 transition. Chromatin immunoprecipitation (ChIP) experiments demonstrated that there was an enrichment of Mcm1 binding at the *CIP1* promoter. This binding was diminished in the *ecb I + II* mutants (Fig. 2e).The binding of Mcm1 peaked at 80 min, the late mitosis stage, after cells released from G1 phase (Supplementary Fig. 1). Together these findings indicate that Mcm1 regulates *CIP1* expression periodically at M/G1 transition through directly binding to the *CIP1* promoter.

**CIP1 expression is induced upon stress.** In natural environments, yeast encounters various stresses. Since *CIP1* is a cell cycle inhibitor, we were curious whether the expression of

*CIP1* could be regulated by stresses. *CIP1* mRNA expression was examined under several stresses including osmotic stress (KCl), oxidative stress ($H_2O_2$), carbon source starvation (low glucose) (Fig. 3a–e), DNA damage (MMS), heat shock (37 °C), and rapamycin treatment (Supplementary Fig. 2a–e). Northern analysis showed that *CIP1* mRNA increased upon the treatment of osmotic stress, oxidative stress, carbon source starvation, and DNA damage. *CIP1* expression reached the peak at 15 min after stress, and then gradually declined. Further investigation indicated that *CIP1* expression increased at 5 min after KCl treatment (Supplementary Fig. 2f). However, spotting assays detected that *cip1* deletion mutants were not sensitive to these stresses (Supplementary Fig. 3a, b), which could indicate redundancy. In any case, these results indicate that *CIP1* may be involved in immediate stress response and cause temporary G1/S delay for adaptation to sudden changes of environment.

**Stress-driven *CIP1* expression depends on Msn2/4.** Functional redundant transcription factors Msn2 and Msn4 can activate the transcription of stress response genes[31–35]. To examine whether the stress-induced *CIP1* expression is controlled by Msn2/4, we monitored the *CIP1* mRNA expression in wild-type and *msn2 msn4* double mutants under stress treatment. *CIP1* mRNA expression was not induced in *msn2 msn4* cells (Fig. 3a–e).

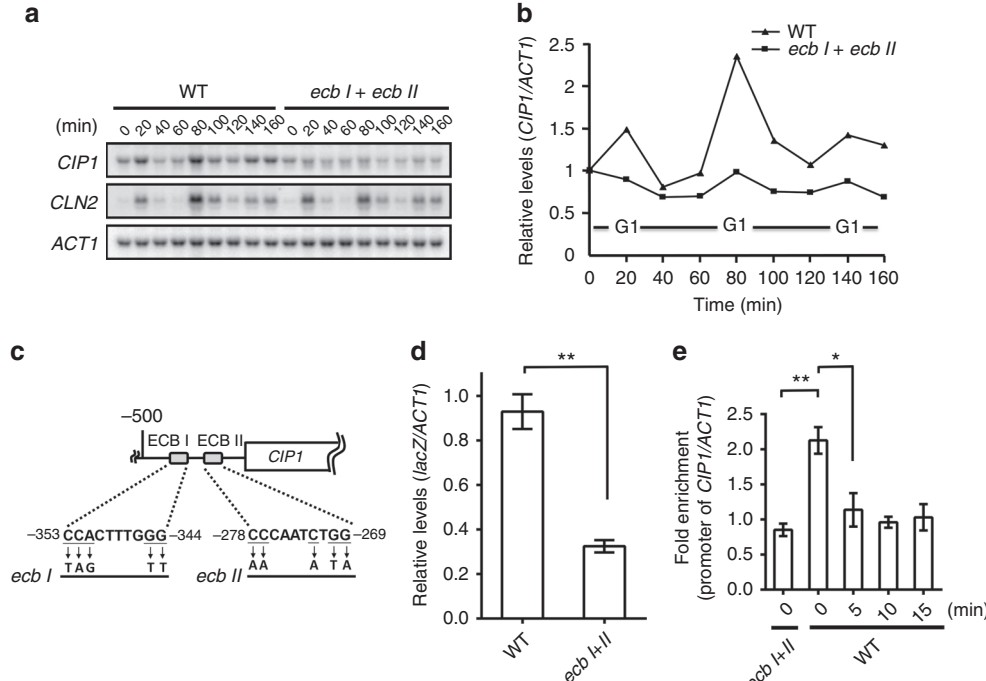

**Fig. 2** *CIP1* expression at M/G1 is mediated by Mcm1 through ECB elements on *CIP1* promoter. **a** After synchronized cells had been released from G1 phase, total RNA was extracted from WT and *ecb I* + *ecb II* strains and analyzed by Northern blotting. **b** The quantitative results of the relative amount of *CIP1* normalized to the internal control, *ACT1*, was shown. **c** Schematic diagram indicating two potential ECB elements on the *CIP1* promoter. The mutation sites that destruct two ECB elements are shown below. **d** Reporter analysis was used to determine the contribution of ECB elements on the promoter activity of *CIP1*. The *CIP1* promoter was placed at the upstream of the *lacZ* gene and then integrated into the yeast genome. To create the *ecb I* + *II* mutant, the ECB elements on the integrating plasmid were mutated before insertion. The *lacZ* expression was normalized to *ACT1*. The values were given as mean ± s.d. ($n = 3$,**$P < 0.01$, Student's *t*-test, two tailed). **e** ChIP analysis of Mcm1 binding at *CIP1* promoter and the times indicated the minutes after 0.5 M KCl treatment. The binding of Mcm1 on *CIP1* promoter was normalized with *U2* background. The values were given as mean ± s.d. ($n = 3$, *$P < 0.05$, **$P < 0.01$, Student's *t*-test, two-tailed)

Furthermore, the ChIP results uncovered an enhanced Msn2 binding to the *CIP1* promoter upon stress (Fig. 3f–h). Interestingly, association of Mcm1 at the *CIP1* promoter decreased under osmostress (Fig. 2e).These results suggest that the stress-induced *CIP1* expression is regulated by direct binding of Msn2/4 to the *CIP1* promoter.

Several regulatory pathways suppress Msn2/4 activity. For example, inhibition of PKA or TOR pathway activates Msn2/4[38–40]. Since *CIP1* is a downstream target of Msn2/4, it might be spontaneously upregulated in the absence of Msn2/4 suppressor even without stresses. *CIP1* expression was slightly increased in RAS, GPA, and TOR pathway blunted cells (Supplementary Fig. 4a). However, these mutants did not display an apparent growth defect upon osmostress (Supplementary Fig. 4b). These results suggest that RAS, GPA, and TOR pathways are relevant to Msn2/4-driven *CIP1* expression. However, since the single abolishment of one of these pathways only slightly increases *CIP1*, it might not be sufficient to display a growth defect.

Upon stress, Msn2 and Msn4 translocate from cytoplasm into nucleus and bind to the STRE[32, 33]. Two potential consensus STRE sites were observed at the upstream of *CIP1* (Fig. 4a). To determine the contribution of these sequences for *CIP1* expression, we generated a chromosomal *lacZ* reporter strain containing mutations on both STRE sites of the *CIP1* promoter. Under stress, the *lacZ* expression was increased, but this upregulation was diminished when both STREs were mutated (Fig. 4b). Electrophoretic mobility shift assay (EMSA) further detected that recombinant Msn2 binds to these STREs directly (Fig. 4c). These results suggested that the STREs contribute to the

transcriptional regulation of *CIP1* promoter. Next, we directly created mutations of the STREs at the endogenous *CIP1* promoter. Stress-induced *CIP1* expression was diminished in *stre I*, *stre II*, and *stre I* + *II* mutant strains (Fig. 4d). The contribution of the two STREs was different: *CIP1* mRNA expression was more significantly reduced in the *stre I* strain and only additively decreased in *stre I* + *II* strain. These results demonstrate that both STREs are critical for stress-driven *CIP1* expression when cells face environmental challenges. ChIP assay revealed that Msn2 directly binds to the STREs at the *CIP1* promoter upon osmostress and this recruitment was inhibited in the *stre I* + *II* cells (Fig. 4e). Together these findings reveal that Msn2/Msn4 facilitate *CIP1* expression under stresses.

**Cip1 is phosphorylated under osmotic stress**. Under osmotic stress, *CIP1* mRNA was upregulated (Fig. 3b,e). We further examined Cip1 protein expression pattern under hyperosmotic stress and found that there was a gel mobility shift of Cip1 under hyperosmotic conditions. This shift was diminished by phosphatase treatment (Supplementary Fig. 5a). We wondered whether the phosphorylation of Cip1 is critical for yeast in adaptation upon osmostress. Initially, by using mass spectrometry (MS) analysis of purified GST-Cip1 from KCl-treated cells (Supplementary Fig. 5b), we identified that T65, T69, and T73 (hereafter called 3T) were phosphorylated under osmostress (Supplementary Fig. 5c). To determine whether the phosphorylation is critical for cells to adapt to the osmostress, we mutated T65, T69, and T73 from threonine to alanine to mimic non-phosphorylated Cip1, or to glutamate in an attempt to

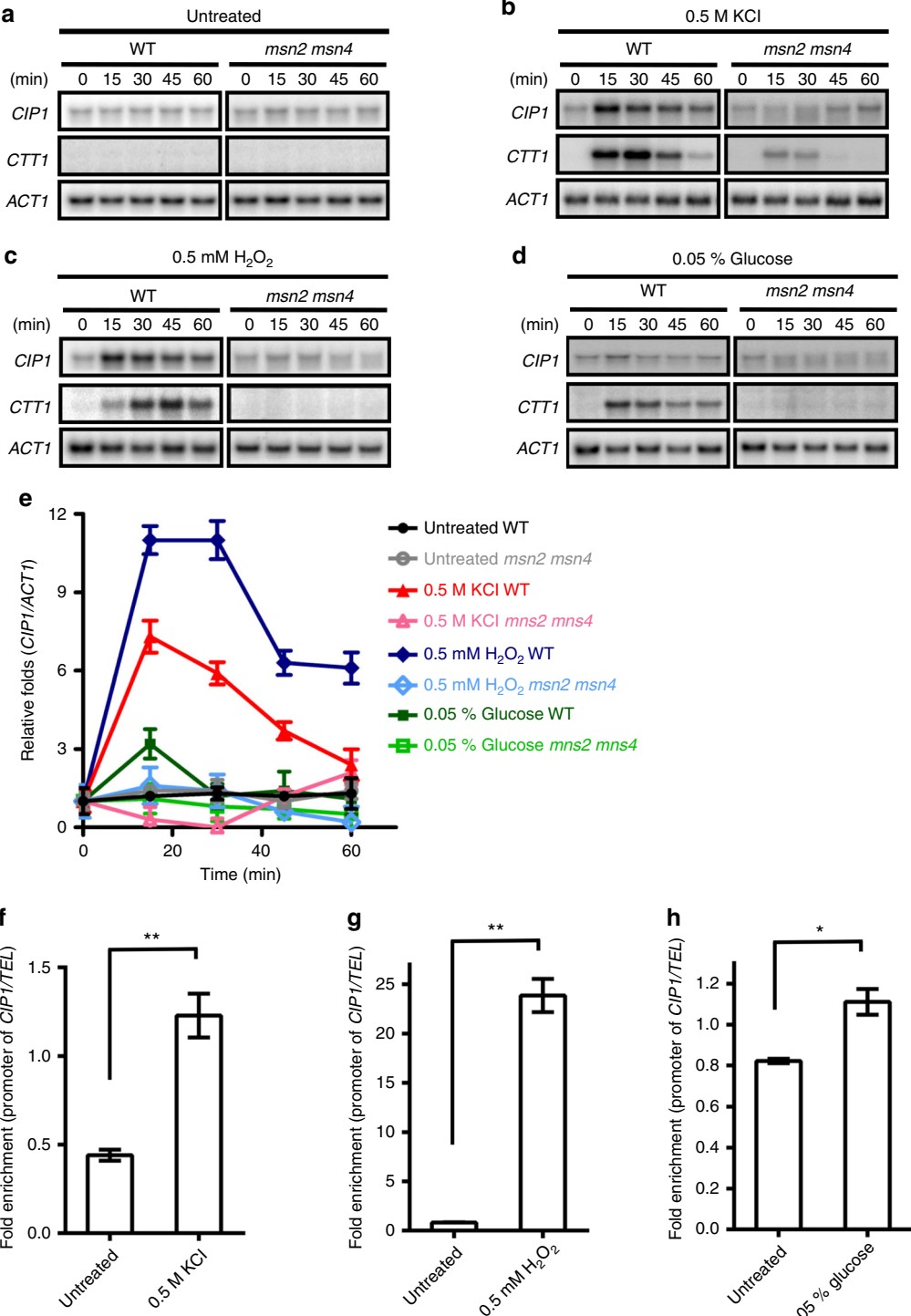

**Fig. 3** *CIP1* expression is induced under several stresses. Yeast cells were treated with various stresses including **a** untreated, **b** osmotic stress (0.5 M KCl), **c** oxidative stress (0.5 mM $H_2O_2$), and **d** carbon source starvation (0.05% glucose). Samples were collected at the indicated time points after stress treatment. Total RNA was analyzed by Northern blotting and hybridized with indicated probes. *CTT1* referred to the positive control of these stresses, and *ACT1* was used as a loading control. **e** The band intensities displayed in the broken-line graph of each panel were quantified using Image J, normalized relative to respective internal controls, and expressed as the ratio of the *CIP1* levels to the time point 0 at the beginning of each stress treatments. The values were given as mean ± s.d. ($n = 3$). mRNA expression was considered to be upregulated when relative fold exceeds 2. **f–h** ChIP analysis of Msn2 binding at the *CIP1* promoter under **f** osmotic stress (0.5 M KCl), **g** oxidative stress (0.5 mM $H_2O_2$), and **h** carbon source starvation (0.05% glucose). The binding of Msn2 at the *CIP1* promoter was normalized by that at the *TEL* sequence. The values are given as mean ± s.d. ($n = 3$, *$P < 0.05$, **$P < 0.01$, Student's *t*-test, two-tailed)

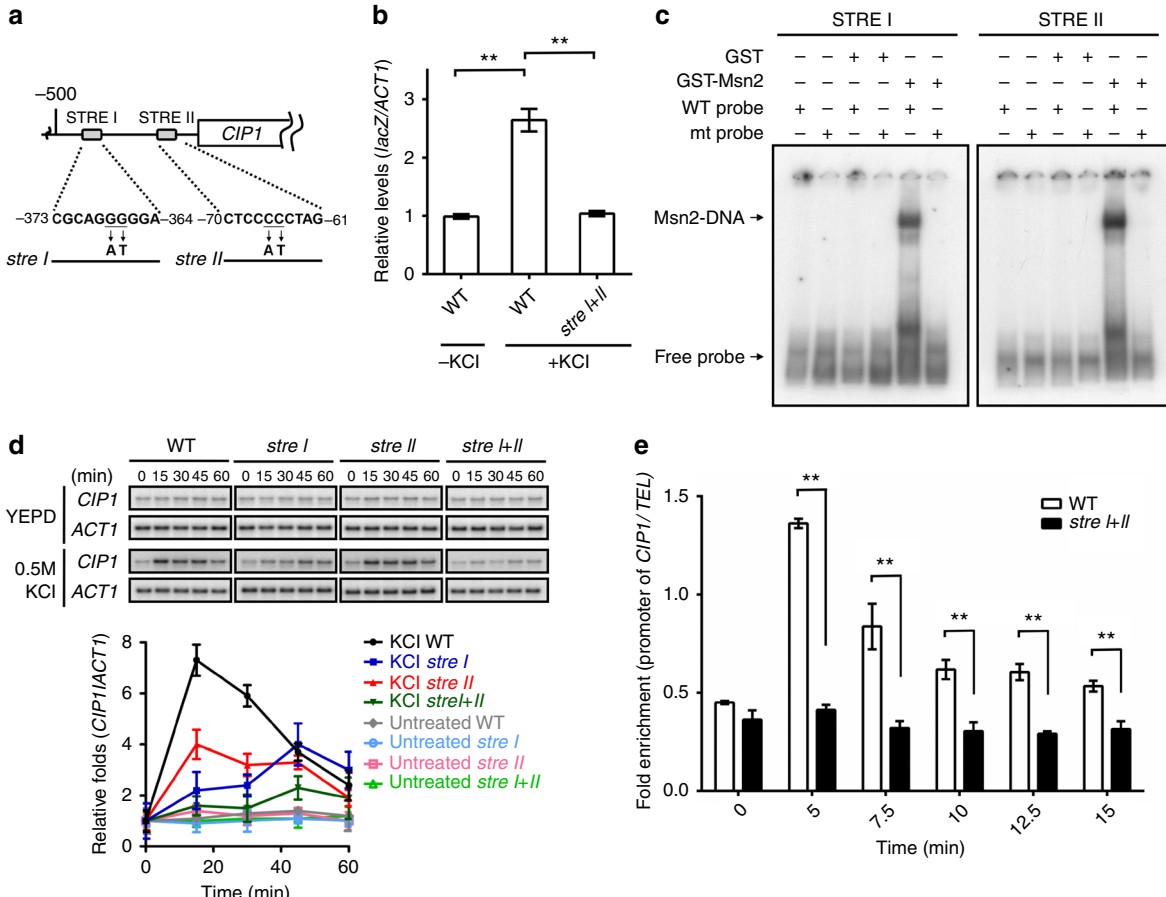

**Fig. 4** Msn2/4 regulate *CIP1* expression through two STRE binding sites. **a** Schematic diagram indicates two putative STREs at the *CIP1* promoter. The mutation sites that destruct STREs are shown below. **b** Reporter analysis was used to determine the contribution of STRE on the promoter activity of *CIP1*. Osmotic stress was induced by 0.5 M KCl. The *lacZ* expression was normalized by *ACT1* expression. The values are given as mean ± s.d. (n = 3,**P < 0.01, Student's t-test, two-tailed). **c** EMSA was performed with STRE I, mutant stre I, STRE II or mutant stre II-radiolabeled double-stranded oligonucleotides. Recombinant GST or GST-Msn2(401–704) proteins were subjected to EMSA. **d** *Top*: Northern blot analysis of *CIP1* expression in WT and STRE mutant strains (*stre I*, *stre II*, and *stre I + II*) under osmotic stress. *Bottom*: The band intensities displayed in the broken-line graph was quantified using Image J, normalized relative to respective internal controls, and expressed as the ratio of the *CIP1* levels to the time point 0 at the beginning of 0.5 M KCl treatments. The values were given as mean ± s.d. (n = 3). **e** ChIP analysis of Msn2 binding at *CIP1* promoter under osmotic stress at indicated times. The binding of Msn2 at the *CIP1* promoter was normalized by the *TEL* background. The values are given as mean ± s.d. (n = 3, **P < 0.01, Student's t-test, two-tailed)

mimic phosphorylated Cip1. The spotting assay indicated that overexpressed *CIP1*, but not *cip1–3TA*, caused growth inhibition (Fig. 5a). Overexpressed Cip1–3TE did not delay cell growth as the wild-type Cip1 (Fig. 5a), implying that the Cip1–3TE mutation may not be sufficient reflection of the wild-type function. Moreover, to identify the contribution of each phosphorylation site on growth, T65, T69, and T73 were double mutated with each other to alanine. Double mutations of T65A and T73A were sufficient to lose the ability of growth inhibition, but not the other combinations (Supplementary Fig. 6a), suggesting that the phosphorylations of T65 and T73 are necessary for Cip1-induced growth inhibition.

**Hog1 phosphorylates Cip1 T65 and T73 under osmotic stress.** Hog1 is the effector protein kinase of the high osmolarity glycerol (HOG) pathway and transiently translocates from the cytosol to nucleus where it controls gene expression during hyperosmoadaption[41, 42]. Under the hyperosmotic condition, Hog1 is essential for cell survival through regulating gene expression, glycerol accumulation, signal fidelity and cell cycle arrest[43, 44].

Since the T65, T69 and T73 of Cip1 were phosphorylated under osmotic stress, we speculated Hog1 as the potential kinase. To determine whether Hog1 directly phosphorylates Cip1, in vitro kinase assay was conducted using purified recombinant Hog1, the Hog1 activator Pbs2 MAPKK, and Cip1. Using Sic1 as a positive control[45], the in vitro kinase assay showed that Hog1 directly phosphorylated Cip1, but not Cip1–3TA. Similar inhibition was observed when a Hog1 inhibitor (SB203580) was included (Fig. 5b), suggesting that Cip1 is a direct target of Hog1. We further generated phosphor-specific antibodies (Supplementary Fig. 6b and c) to detect the phosphorylation states of Cip1 under osmotic stress. A prompt phosphorylation of Cip1 was observed within 5 min after hyperosmotic stress, and the phosphorylations were then gradually decreased at the following time points (Fig. 5c). In *hog1* cells, Cip1 T65 and T73 were not phosphorylated under osmotic stress (Fig. 5c), suggesting that these phosphorylations require Hog1. Together, these results demonstrate that Hog1 directly phosphorylates Cip1 upon hyperosmotic stress. The status of Cip1 phosphorylation during cell cycle progression was also monitored. There was no detectable Cip1 T65 and T73 phosphorylation during the cell

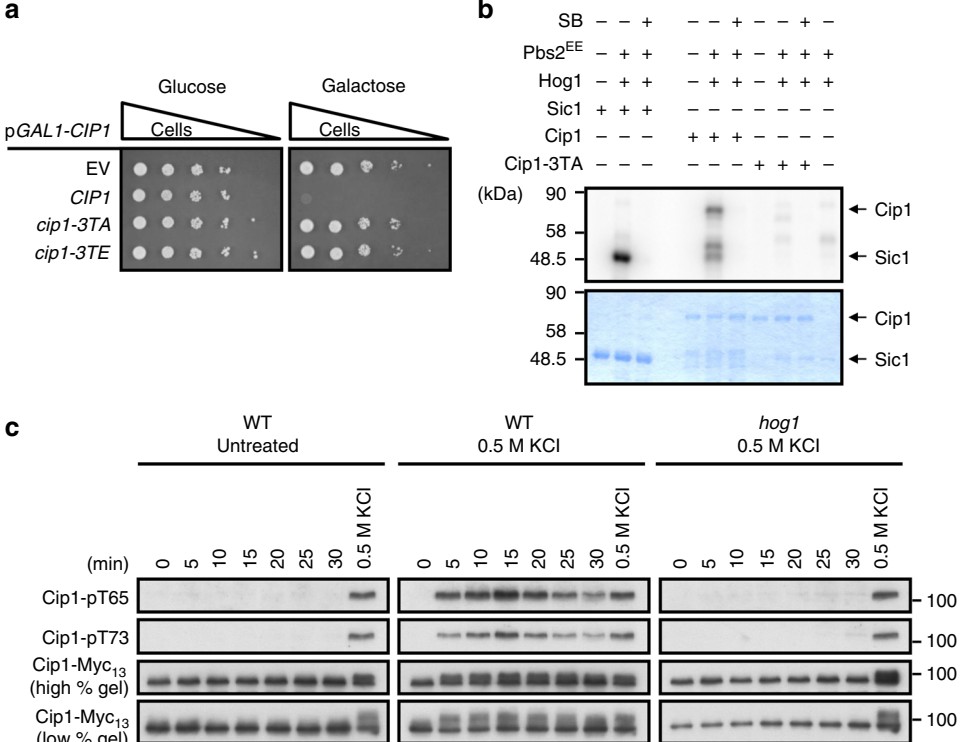

**Fig. 5** Cip1 is phosphorylated by Hog1 kinase directly under osmotic stress. **a** Isogenic WT strains bearing empty vector, *GAL1-CIP1*, *GAL1-cip1-3TA*, or *GAL1-cip1-3TE* plasmid were spotted in 10-fold diluted equal number of yeast on 2% glucose and 2% galactose plates. **b** Recombinant GST-Sic1, GST-Cip1, and GST-Cip1-3TA were employed as substrates for *in vitro* kinase assay. GST-Hog1 was activated by incubation with GST-Pbs2EE for 30 min at 30 °C. Activated Hog1 was added to the substrate and incubated at 30 °C for 30 min in the presence of $^{32}$P-ATP. Proteins were resolved by SDS–PAGE and stained with Coomassie blue (*bottom*), and phosphorylated proteins were detected by autoradiography (*top*). Sic1 was included as a positive control; SB203580 was used to selectively inactivate Hog1. **c** Cip1 phosphorylations in WT cells, WT cells treated with 0.5 M KCl, and *hog1* cells treated with 0.5 M KCl were detected by phosphor-specific antibodies, pT65 and pT73, and total Cip1 was served as a loading control. The mobility shift of phosphorylated Cip1 was shown in low percentage gel

cycle (Supplementary Fig. 6d). A previous study showed that Cdk1–Clb complex phosphorylates Cip1[36] and both Cdk1 and Hog1 share the same consensus phosphorylation sites (S/T-P). To further understand whether the phosphorylations of T65 and T73 were cell cycle-dependent, the *cdc28-as1* strain was treated with 4-amino-1-tert-butyl-3-(1′-naphthylmethyl) pyrazolo [3,4-d] pyrimidine(1-NM PP1) to block Cdk1 activity under osmotic stress[46]. Inhibition of Cdk1 activity displayed no influence on osmostress-induced Cip1 phosphorylations (Supplementary Fig. 6e). These results indicate that Cip1–3T phosphorylation is only induced under osmostress, but not in regular mitotic cycle.

**Cip1 contributes to osmostress-induced transitory G1 delay.** Osmotic stress causes transient G1 delay of cells, which leads to retarded budding formation[7]. *CIP1* mRNA is upregulated upon hyperosmotic stress (Fig. 3b,e) and Cip1 overexpression leads to cell cycle arrest (Fig. 1b). To determine whether Cip1 contributes to transient G1 arrest under hyperosmotic conditions, the fraction of budded cells of *cip1* strains in exponentially growing cultures was measured. Consistent with previous findings[7], the lowest budding index of wild-type cells was detected at 40 min after KCl treatment. The budding index of the *cip1* cells at 40 min was higher than that of wild-type cells (Fig. 6a), suggesting that Cip1 contributes to osmotic stress-induced transitory cell cycle arrest. The budding index of *cip1–3TA* cells was similar to that of *cip1* cells; both are higher than wild-type

cells, suggesting that Hog1 controls Cip1-dependent cell cycle delay under osmotic stress.

Sic1 was reported as a CKI that causes G1 arrest under osmotic stress[45]. To evaluate the contribution of Sic1 and Cip1 in response to osmostress, the budding indexes of *sic1* and *sic1 cip1* strains were also monitored. Deletion of *SIC1* gave rise to more budded cells under untreated condition (Fig. 6a, time 0), suggesting that *SIC1* plays an essential role in G1 in the regular mitotic cycle. At 40 min after KCl treatment, the change of budding index of *sic1* cells (77.6 to 60.3%) was almost the same as that of *cip1* cells (65.8 to 47.8%; Fig. 6a). Surprisingly, the budding index was much higher in *sic1 cip1* double mutants than either *sic1* or *cip1* single mutants (Fig. 6a), suggesting a redundant role of these CKIs in the acceleration of S-phase entry. We observed that α-factor treatment downregulates Cip1 expression (Supplementary Fig. 6d). To further investigate the biological functions of Cip1 but avoid the disturbance of α-factor treatment, we elutriated wild-type, *cip1*, *sic1*, and *cip1 sic1* strains to analyze the cell cycle progression under osmotic stress of newborn cells. The FACS analysis showed no difference between wild-type and *cip1* cells, while the loss of *CIP1* in *sic1* cells further accelerated cell cycle progression at 120 min after osmotic stress treatment (Fig. 6b). Also, *CIP1* single deleted cells did not exhibit a stress-sensitive phenotype compared to wild-type cells; however, deletion of *CIP1* further augmented the stress-sensitive phenotype in *sic1* cells (Fig. 6c; Supplementary Fig. 3b). These results revealed that Cip1 collaborates with Sic1 in osmotic stress-induced G1 cell cycle arrest, implying that Sic1 and Cip1

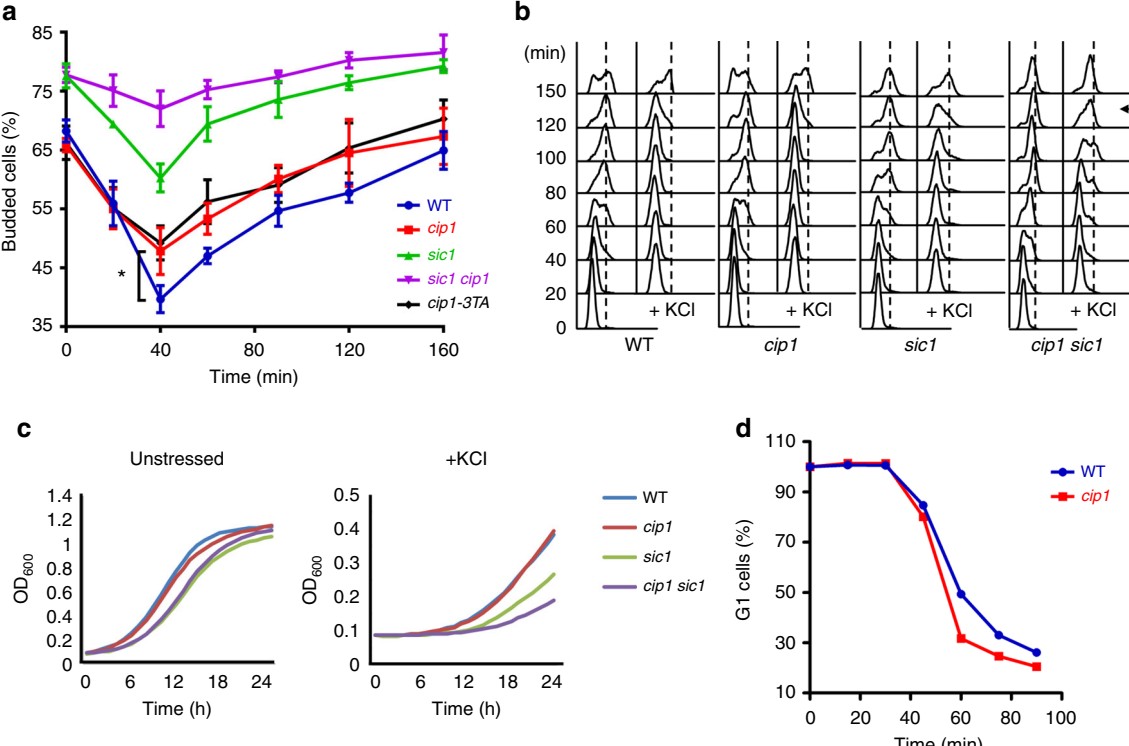

**Fig. 6** The phosphorylation of Cip1 is responsible for hyperosmotic stress-induced transitory G1 delay. **a** Budding index of WT, *cip1*, *cip1-3TA*, *sic1*, and *sic1 cip1* strains under treatment of 0.5 M KCl. The percentage of budded cells were given as mean ± s.d. ($n = 3$, *$P < 0.05$, Student's *t*-test, two-tailed). **b** WT, *cip1*, *sic1* and *cip1 sic1* strains were elutriated in YEP, raffinose. Newborn cells were collected and recovered in YEPD at 25 °C. At time 0, cultures were subjected to or not to osmostress (0.5 M KCl). Cell cycle progression was monitored by FACS analysis for 150 min after stress. An arrow indicates the timing when the G1-S transition speeds up in *cip1 sic1* cells. **c** Exponentially growing cells were diluted to $OD_{600} = 0.1$ and grown at 30 °C in minimal media. Osmosensitivity was tested in the presence of 1.2 M KCl. OD measurements were estimated every hour for 25 h in a 96-well plate using Synergy H1 Multi-Mode Reader. **d** In WT and *cip1* strains, α-factor synchronized G1 cells were released in 0.5 M KCl YEPD medium at 24 °C. At 15 min intervals, samples were collected, and α-factor/nocodazole trap assay was performed. More than 300 cells showing mating projections (G1 cells) or buds (post-G1 cells) were counted for each time point, and the experiment was repeated three times

are functionally redundant proteins when cells encounter environmental challenges.

Although the results of budding index indicate that Cip1 contributes to osmotic stress-induced transitory cell cycle arrest, the regular FACS analysis of wild-type and *cip1* cells showed no obvious difference of cell cycle progression under the treatment of osmotic stress (Fig. 6b; Supplementary Fig. 7a). To further distinguish early S phase from G1 arrest, which might be insensitive to FACS, we analyzed the wild-type and *cip1* cells using an α-factor/nocodazole trap assay. G1 synchronized wild-type and *cip1* cells were released in medium containing 0.5 M KCl. At 15 min intervals, cells were transferred to trapping medium (α-factor and nocodazole) and incubated for 90 min. Those cells that remain in G1 were sensitive to α-factor and form mating projections; in contrast to any cells that left G1 passing START were arrested in nocodazole at G2/M as large budded cells. Performing this assay under osmotic stress revealed that *cip1* cells exited G1 earlier than wild-type cells (Fig. 6d). Altogether, these results indicate that Cip1 regulates osmotic stress-induced G1 delay through Hog1-mediated phosphorylation.

**Cip1 phosphorylation may enhance the binding to Cdk1–Clns.** Cip1-dependent slow growth can be partially rescued through simultaneous induction of G1 cyclin (Fig. 1d), implying that Cip1-mediated G1 arrest is probably due to the inhibitory

targeting of Cip1 to Cdk1–G1 cyclin complexes. A previous study found that Cip1 physically interacts with the Cdk1–Cln2 complex[36]. To investigate the physical interaction of Cip1 with the three Cdk1–G1 cyclin complexes, we performed co-immunoprecipitation to determine their associations. The Myc-tagged Cip1 and HA₃-tagged Cln1, Cln2, or Cln3 proteins were overexpressed under the *GAL1* promoter. Under osmotic stress, Cip1 exhibited higher binding affinity to each of the G1 cyclins (Fig. 7a). These interactions decreased in the *cip1-3TA* mutant (Fig. 7b). Collectively, these data suggest that Cip1 can interact with all Cdk1–G1 cyclin complexes and the Hog1-mediated phosphorylation might strengthen the binding affinity (Fig. 7c).

**Cip1 impedes G1 through inhibiting Cdk1–Cln3 activity.** Among three G1 cyclins, Cln3 is the most upstream regulatory G1 cyclin before the START. The Cdk1–Cln3 phosphorylates Whi5 for the activation of SBF, which induces specific G1–S transition genes[16]. We therefore speculated that Cdk1–Cln3 may be the most upstream target of Cip1 at early G1. To confirm whether Cip1 delays early G1 phase through inhibiting Cdk1–Cln3, the yeast strains bearing the empty vector or *GAL1–CIP1* overexpressing plasmid were synchronized at G1 and Whi5 phosphorylation was monitored after release (Supplementary Fig. 7b). In cells carrying an empty vector, phosphorylated Whi5 appeared and peaked at 75 min after releasing (Fig. 7d).

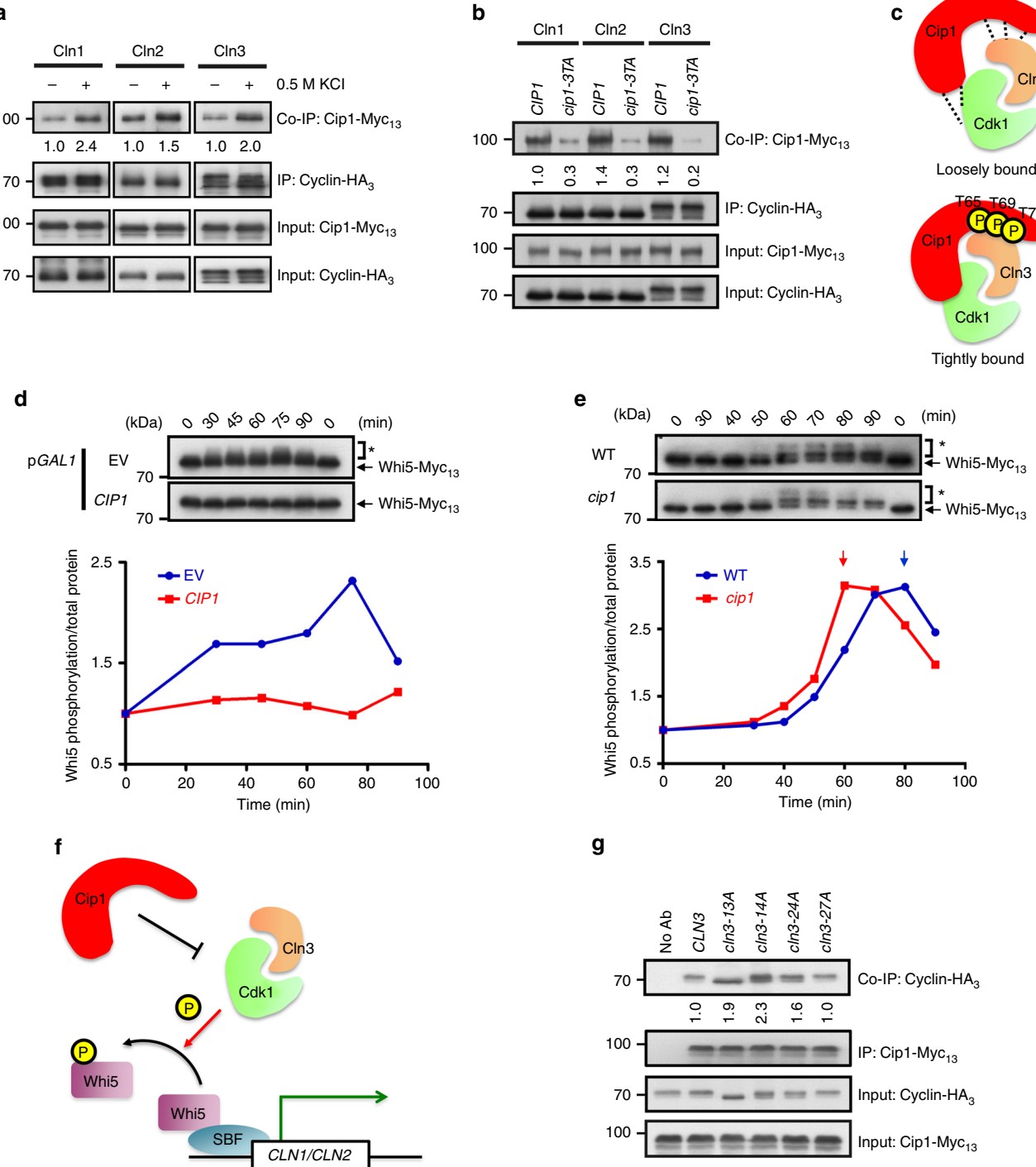

However, this phenotype was not observed in *CIP1*-over-expressing cells. Under osmotic stress, yeast cells arrest in G1 transiently[7]. To determine whether Cip1 could transiently inhibit Cdk1–Cln3 activity under hyperosmotic stress, the timing of appearance of phosphorylated Whi5 in wild-type and *cip1* cells were monitored. α-factor released cell cycle progression was analyzed by FACS analysis (Supplementary Fig. 7c). After KCl treatment, the Whi5 phosphorylation in *cip1* cells peaked at 60 min, which was 20 min faster than that in wild-type cells (Fig. 7e), suggesting that Cip1 inhibits Cdk1–Cln3 activity and thereby prevents Whi5 phosphorylation. The unphosphorylated Whi5 suppresses SBF and causes cell cycle arrest (Fig. 7f).

**Cdk1–Cln3 interaction prevents Cip1–Cln3 binding.** We next asked whether the Cdk1–Cln3 interaction is crucial for Cip1 function. Several sites on Cln3 are critical for its interaction with Cdk1[47]. To this end, the interactions between Cip1 and wild-type Cln3, three Cln3 mutants, Cln3-13A (K163D166A), Cln3-14A (D166R170A), and Cln3-24A (K357K359A), which are defective in Cdk1 interaction, and a non-related control mutant, Cln3-27A (E386E387AR390A), which maintains the Cdk1–Cln3 interaction, were examined. In contrast to wild-type Cln3 and a non-related control (Cln3-27A), the interaction between Cip1 and Cln3 increased in the Cdk1 interaction-defective *cln3* mutants, *cln3-13A*, *cln3-14A*, and *cln3-24A* (Fig. 7g). These results revealed

that Cip1 exhibits a stronger association with Cdk1 interaction-defective Cln3, implying an enhanced binding of Cip1 to the free form of Cln3. This Cip1–Cln3 binding might block Cdk1 to form a complex with Cln3 and therefore inhibit the kinase activity.

**Cip1 localizes to the nucleus upon osmotic stress**. Our findings indicate that Cip1 can interact with all Cdk1–G1 cyclin complexes to inhibit cell cycle progression and the Hog1-mediated phosphorylation on Cip1 strengthens the Cip1–Cdk1–Cln interaction. Under osmotic shock, Hog1 translocates transiently to the nucleus[48]. To investigate the localization of Cip1 under osmotic stress, a C-terminal GFP-fused Cip1 was overexpressed in wild-type and *hog1* cells. Fluorescence microscope examination revealed that Cip1 was localized throughout the cytoplasm and nucleus in unstressed cells. When cells were exposed to osmotic shock, the fluorescence signal of Cip1 accumulated within the nucleus. In *hog1* cells, Cip1 was still able to accumulate in the nucleus under osmotic stress (Supplementary Fig. 8). These data suggest that Cip1 localizes to the nucleus after osmotic stress.

**Human p21 expression stimulates G1 arrest in yeast cells**. Cip1 is induced by stress and activated by Hog1 for G1 arrest. The mammalian homolog of Hog1, p38, also phosphorylates downstream stress-induced p21. Since Cip1 contains a conserved CDK-binding motif found in human p21[36], it drew our attention that Cip1 might be a homologue of human p21. The spotting assay showed that overexpression of p21 resulted in inhibition of yeast growth. This growth retardation was caused by G1 arrest as determined by FACS analysis (Supplementary Fig. 9). The functional similarity of p21 to Cip1 implies that Cip1 may be a homolog to human p21.

**Discussion**
Cip1 was recently characterized as a new G1 CKI which associates with Cdk1 and Cln2[36]. However, its detailed regulatory mechanisms to control cell cycle progression are poorly understood. Our studies uncovered that Cip1 not only inhibits Cdk1–Cln2 but all Cdk1–Cln complexes to arrest G1 progression. *CIP1* is specifically transcribed from M/G1 to late G1, and its expression is modulated by the upstream activating sequence in an Mcm1-dependent manner *in vivo* at M/G1. In the previous study, *CIP1* was also found to be regulated by Mbp1, a G1 transcription factor. We infer that Mcm1 binds to Cip1 promoter at M/G1 stage and the early expressed Cip1 mainly regulates Cdk1–Cln3 at early G1. Mbp1 subsequently induces *CIP1* expression, and the Cdk1–Cln1/Cln2 are the primary targets of Cip1 at late G1 (Supplementary Fig. 10).

Our studies also demonstrated that *CIP1* is upregulated by various stresses, which is mediated by the key regulators of the stress-responsive genes, Msn2/4. The STREs at the *CIP1* promoter are essential for the binding of Msn2/4 to promote *CIP1* expression under stress. In wild environments, organisms require the proper response and protective mechanism to adapt to stress. There are many stress response genes expressed to help cells to overcome the external challenge. Cip1 may be the stress response gene to regulate G1/S transition. Under hyperosmotic condition, *cip1* cells escape from transitory G1 arrest and display higher budding index. α-factor/nocodazole assay also reveals that *cip1* cells exit G1 faster under osmotic stress. Cells need time to activate a particular set of genes for building up proper cellular responses to adapt to stress and recover from damage. Cip1 may be the regulatory protein responsible for the G1 arrest, which ensures cells to make the appropriate response to overcome the challenge before S phase initiation. Even though *CIP1* is induced in response to multiple stresses, *CIP1*-deleted cells are not sensitive to these stresses. It is likely that Cip1 causes transient G1/S delay, in collaboration with Sic1, for adaptation to sudden environmental changes, but later on the delay might be recovered in the following cell cycle stages. Furthermore, Hog1-mediated Whi5 and Msa1 phosphorylations also inhibit G1 cyclins expression to control adequate passage through START upon osmostress[17]. Therefore, multiple regulatory pathways at G1/S might accumulatively delay G1 for cells to cope with stress.

The osmotic stress responses are mostly regulated by the SAPK signaling pathway[49]. Hog1 is the effector kinase of this pathway, which controls gene expression during hyperosmoadaption[42–44]. Hog1 phosphorylates Cip1 T65, T69, and T73 upon hyperosmotic stress. However, the phosphorylation of Cip1 could only be detected from the TCA-treated total lysates, due to the significant instability of phosphorylated Cip1 immediately after cell breakage. The Hog1-mediated phosphorylation might increase the interaction with Cdk1–G1 cyclins to induce transitory cell cycle arrest. All these findings establish that Cip1-mediated cell cycle arrest occurs through the inhibitory targeting to Cdk1–G1 cyclins (Fig. 8). Cip1–3TA loses the inhibitory phenotype of Cip1; however, the phospho-mimetic Cip1 mutant does not restore the wild-type function. It is not unusual that phosphomimetic mutants cannot completely simulate the structures of phospho-serine and phospho-threonine, given the fact that the negative charge introduced by glutamate or aspartate substitution (−1) does not match with that of the phosphorylated residue (−2) at physiological pH[50]. The inhibitory association of Cip1 with the most upstream regulatory Cdk1–G1 cyclin, Cdk1–Cln3, inhibits Whi5 phosphorylation. Without CDK-mediated phosphorylation, Whi5 inactivates SBF and results in temporary G1/S delay.

**Fig. 7** Cip1 regulates G1/S progression by inhibiting Cdk1–Cln3 complex activity. **a** Co-immunoprecipitation indicated that osmotic stress augments the Cip1–G1 cyclin interactions. Immunoprecipitation was conducted using an anti-HA antibody to pull-down *GAL1* promoter-driven cyclins. The levels of signal compared with that of the untreated cells are shown below. **b** Co-immunoprecipitation indicated that Cip1 phosphorylation site mutation impairs the Cip1–G1 cyclin interactions. **c** A cartoon describes the function of phosphorylation of Cip1. The phosphorylation of Cip1 on T65, T69, and T73 increases the binding affinity between Cip1 and Cdk1–Cln3 complex. **d** The Myc$_{13}$-tagged *WHI5* strain harboring empty vector, *GAL1-CIP1*, or *GAL1-cip1-3TA* was synchronized at G1 by α-factor in 2% raffinose. Cip1 was induced in the present of 2% galactose. The α-factor was subsequently removed from the cultures and samples were collected at 15-min intervals for 90 min at 24 °C. Whi5 was detected by western blotting using an antibody against Myc, and the phosphorylated Whi5 was indicated by an asterisk (*). The relative band intensities of phosphorylated Whi5 to total Whi5 was quantified and expressed as the ratio of the phosphorylated Whi5 levels to the time point 0. **e** The chromosomal *WHI5* was tagged with Myc$_{13}$ in WT and *cip1* cells. Cells were synchronized at G1 by α-factor and released into 0.5 M KCl YEPD. Cells were collected at 10-min intervals for 90 min at 24 °C. Whi5 was analyzed by western blotting as **d**. **f** A cartoon describes the inhibitory pathway of Cip1 to Cdk1–Cln3 complex. The interaction between Cip1 and Cdk1–Cln3 complex inhibits the kinase activity of the Cdk1–Cln3 complex. The inhibition blocks Whi5 phosphorylation, which prevents SBF to transcribe G1/S genes. **g** Co-immunoprecipitation between Cip1 and Cln3, Cln3-13A, Cln3-14A, Cln3-24A and Cln3-27A. Immunoprecipitation was conducted using an anti-Myc antibody to pull-down *GAL1* promoter-driven Cip1. The band intensities displayed below depict quantification of the relative amount of Cln3 proteins that co-immunoprecipitate with Cip1

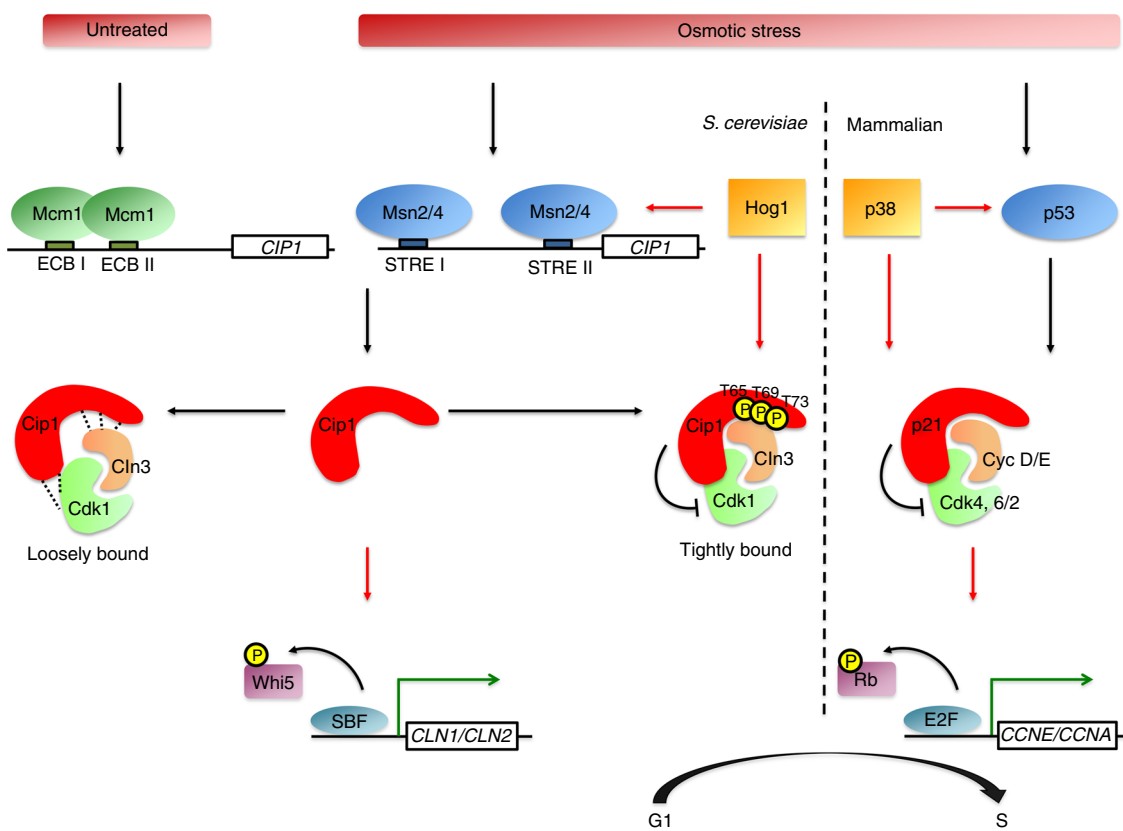

**Fig. 8** The Cip1-involved regulatory pathways of G1/S transition controlled by p38/Hog1 SAPKs in mammal/yeast upon osmotic stress. At M/G1 transition, Mcm1 binds at ECB element on *CIP1* promoter and promotes the transcriptional expression of *CIP1*. Upon osmotic stress, active p38 and Hog1 SAPKs phosphorylate p53 and Msn2/4, respectively. In budding yeast, expression of Cip1 is induced by Msn2/4 and expressed Cip1 is phosphorylated at T65, T69, and T73 by Hog1. The phosphorylation of Cip1 strengthens the binding affinity with the Cdk1–Cln3 complex. The interaction between Cip1 and Cdk1–Cln3 complex inhibits the Cdk1–Cln3 kinase activity and prevents Cdk1–Cln3-complex-dependent Whi5 phosphorylation. Non-phosphorylated Whi5 binds to SBF, blocks G1/S genes transcription, and delays G1/S transition. In mammalian cells, the CDK inhibitor p21 is induced by activated p53 which leads to G1 arrest through inhibition of Cdk4, 6/2-cyclin D/E complexes

Based on the structures and CDK targets, CKIs are divided into two families. The canonical Kip family CKIs bind both cyclin and CDK to block the kinase activity. In contrast, the INK4 family proteins specifically bind CDK monomers, which twists the CDK to disrupt cyclin binding and kinase activity. Over-expressed G1 cyclins can partially overcome the Cip1 block of these cells in G1. On the other hand, Cip1 displays greater association with free form Cln3. This binding on Cln3 may impair Cdk1 to form a complex with Cln3. Hence, Cip1 is not considered as a canonical CKI.

In a normal cell cycle, Cip1, like Sic1, is expressed at the end of mitosis and persists throughout the G1 phase. However, unlike *SIC1* whose expression is not altered under stresses; *CIP1* is induced immediately by stresses, including sorbitol-induced hyperosmotic stress, carbon source starvation, oxidative stress and MMS-triggered damage[51–54]. Sic1 was previously reported to cause G1 arrest under hyperosmotic stress[45]. Surprisingly, the budding index was additively higher in *sic1 cip1* double mutants and loss of *CIP1* in *sic1* cells further accelerates cell cycle progression after osmotic shock. Thus, Sic1 and Cip1 are functionally redundant CKIs to prevent cells from premature DNA replication when cells encounter environmental challenges. Upon the appearance of environmental stimuli, cells need a swift response to go through the crisis. Deletion of *CIP1* further aggravates the stress-sensitive phenotype in *sic1* cells. We believe that *CIP1* is a stress response gene and its expression slows down G1/S transition to help cells overcome the challenges. Moreover,

Cln overexpression only partially suppresses the Cip1 over-expression effects, suggesting a possibility that Cip1 may also be a Cdk–Clb inhibitor to prevent S phase entry under genotoxic conditions.

In mammalian cells, general stresses activate the SAPK p38[55]. p38 phosphorylates downstream targets which include CKIs p21, p27, and p57 to delay cell cycle progression[56–58]. A recent study revealed that osmotic stress causes a strong G1 delay[56, 59]. Inactivation of p38 abolishes osmostress-induced cell-cycle delay and results in decreased cell viability[56, 59]. The tumor suppressor protein p53 and its downstream target p21 are the main factors to repress cell cycle under stress[60]. p21 blocks cell cycle and suppresses unwanted cell division[61]. The circuitry controlling this response is conserved from humans to yeast, but no stress-induced negative regulators has been found in yeast before. In yeast, G1 progression upon osmotic stress is contributed by the Hog1 SAPK to phosphorylate Sic1 which inhibits Cdk1–Clb activation and promotes Clbs degradation[45, 62], but the expression of Sic1 is not regulated by stress. Interestingly, Cip1 is also a substrate of Hog1 and it is stress-induced to slow down cell cycle. Furthermore, Cip1 contains a conserved match to the CDK-binding motif found in human p21[36]. The function, the SAPK-mediated regulation, and the stress-induced character of Cip1 are partly analogous to those of mammalian p21 in preventing cell cycle entry[56–58]. Moreover, yeast Msn2/Msn4 may act as mammalian p53 in induction of CKI expression to inhibit cell cycle progression under several stresses (Fig. 8). These

observations extend the parallels between yeast and metazoans at the repression of the cell cycle, and raise the intriguing possibility that the yeast Cip1 may be the primordial one, and the mammalian p21 may be a relative latecomer to the system for stress response.

## Methods

**Yeast strains and plasmids**. All yeast strains used in this study were derived from W303 (*MATa/α ura3-52/ura3-52 trp1Δ2/ trp1Δ2 leu2-3_112/ leu2-3_112 his3-11/ his3-11 ade2-1 can1-100/ ade2-1 can1-100*), BY4743 (*MATa/α his3Δ1/his3Δ1 leu2Δ0/leu2Δ0 LYS2/lys2Δ0 met15Δ0/MET15 ura3Δ0/ura3Δ0*), BJ2168 (*MATa prc1-407 prb1-1122 pep4-3 leu2 trp1 ura 3-52*), CHY125 (*MATa ade2-1 ade3::hisG ura3-1 his3-12,15 trp1-1 leu2-3112 can1-100*) and YM4271 (*MATaura3-52 his3-Δ200 ade2-101 ade5 lys2-801 leu2-3112 trp1-901 tyr1-501 gal4Δgal80Δade5:: hisG*). The deletion strains *bar1*, *far1*, and *sic1* are from deletion library (Invitrogen). Mutation strain *sic1::LEU2*, *cip1::KanMX4*, and *sic1::LEU2cip1::KanMX4* used in budding index assay were generated by introducing mutation cassette into the isogenic BY4742 background. Double mutation strain *msn2::KanMX4 msn4::KanMX4* in BY4742 was selected by tetrad dissection. The tagged strains were created by integration of the Myc$_{13}$ tag in-frame downstream of specific genes in the genome of BY4741 or W303a *cdc28 as-1*. The pYESL plasmid contains the p*GAL1-CIP1* plasmid, in which the *URA3* of pYES2 plasmid (Invitrogen) was destroyed by the *LEU2* marker, bearing full-length *CIP1* with C-terminal MYC$_{13}$ tag. The N-terminal GST-tagged Cip1 and p21were expressed from pEGKT plasmid which contains the *GAL1* promoter. The p*GAL1-CLN1*, *-CLN2*, and *-CLN3* plasmids are C-terminal tagged with HA$_3$ in the pYES2 plasmid. For the reporter assays, the 500 bp upstream of the translation start site of *CIP1* was PCR-amplified and cloned into p*lacZ*i (Clontech). Putative ECB elements or STRE sequences in the p*P$_{CIP}$-lacZi* reporter plasmid were mutated by site-directed mutagenesis (Invitrogen). Each reporter was introduced into the YM4271 strain as manufacture described (Clontech). All yeast strains, constructs, and primer sequences used in this study are mentioned in Supplementary Tables 1–3, respectively.

**RNA preparation and Northern blotting**. For total RNA extraction, 5 OD$_{600}$ cells were lysed in 1 ml TRIzol solution (Invitrogen) with glass beads and vigorously vortexed for 5 min at room temperature. The following manipulations were conducted as described by the manufacturer. For Northern analysis, 5 μg of total RNA was separated on a 1.2% agarose gel containing 2.1% formaldehyde in MOPS/formaldehyde buffer. RNA was transferred to a nylon membrane (Perkin-Elmer) and hybridized with [α-$^{32}$P] dCTP-labeled probes for *CIP1*, *CLN2*, *CLN3*, *CTT1*, *RNR1*, *CIT2*, and *ACT1*. Results were analyzed with PhosphorImager and quantitated with ImageJ software. All uncropped Northern blots can be found in Supplementary Fig. 11.

**Cell growth media and condition for stress response**. The yeast media used were rich medium (YEP, 1% yeast extract, 2% peptone) or SC medium containing 2% glucose, 2% raffinose, or 2% galactose as indicated. For Northern blot analysis of cells treated under stress, the overnight culture was grown to early log phase in YEPD at 23 °C. Five OD$_{600}$ cells were collected at 23 °C for each indicated time points after exposing to stresses. For carbon source starvation, the glucose concentration was changed from 2 to 0.05%. Osmotic stress was induced by 0.5 M KCl. Heat shock was performed by changing temperature from 23 to 37 °C. The untreated control of rapamycin was supplied with the same volume of dimethyl sulfoxide.

**EMSA**. The pGEX4T-1-*msn2(401–704)* plasmid was kindly provided by Dr Ji-Sook Hahn. The GST-Msn2(401–704) protein containing the DNA binding domain of Msn2 was expressed in BL21(DE3) and purified[63]. EMSA was performed as previously described[64]. Briefly, 100 ng recombinant proteins were used. $^{32}$P-end-labeled double stranded synthetic oligonucleotides containing the STRE I or STRE II were added. The mutant *stre I* and *stre II* oligonucleotides were used as negative controls. Following electrophoresis on a 4% polyacrylamide nondenaturing gel, the gel was dried and subjected to autoradiography.

**RNA purification and quantitative reverse transcription PCR**. RNA was extracted using TRIzol reagent (Invitrogen). cDNA was synthesized using Thermo Scientific Maxima First Strand cDNA Synthesis (Thermo). Quantitative reverse transcription PCR(qRT-PCR) was performed on a Biorad CFX Connect Real-Time PCR Detection System. All primer sequences for PCR are listed in Supplementary Table 3.

**ChIP analysis**. ChIP analysis was performed as described[65]. In brief, cells were grown as indicated conditions. After crosslinking in 1% formaldehyde, cells were lysed and sonicated. Immunoprecipitations were carried out with an anti-Mcm1 antibody (kindly provided by George Sprague) and anti-Msn2 antibody (Santa Cruz, y-300) against Mcm1 and Msn2, respectively. Both an aliquot of sonicated cleared extract (input) and the immunoprecipitated materials were de-crosslinked

in Tris/EDTA buffer (TE) plus 1% SDS for at least 8 h at 65 °C. Quantification of immunoprecipitated DNA was obtained by qRT-PCR using SYBR Green detection (Kappa) on an Applied Biosystems HT7500 machine and software. After normalized to the background control (promoter of *U2* or *TEL*), the results were expressed as relative fold enrichment binding level on *CIP1* promoter. Primers used in this study are listed in the Supplementary Table 3. The data are presented as the mean of at least three synchronies plus or minus s.d.

**α-Factor/nocodazole trap assay**. The α-factor/nocodazole trap assay was performed as described[66]. Briefly, after cells were synchronized at G1 by α-factor, WT and *cip1* cells were released in 0.5 M KCl YEPD medium at 24 °C. At 15 min intervals, cells were collected, washed once by YEPD, resuspended in 0.5 ml fresh YEPD combined with 0.5 ml trapping medium (10 μM α-factor, 30 μg ml$^{-1}$nocodazole), and incubated at 24 °C for 90 min. Samples were fixed in 70% ethanol, sonicated, and examined by phase microscope to count more than 300 cells showing shmoos (G1 cells) or buds (post-G1 cells). The experiment was repeated three times, and representative results are presented.

**Budding index under osmotic stress**. Wild-type, *sic1*, *cip1*, *cip1-3TA*, or *sic1 cip1* strains were grown to exponential phase in YEPD. After treated with 0.5 M KCl, cells were fixed in 70% ethanol and sonicated. The budding index is defined as the percentage of the whole population carrying a bud, and a total of 400 cells were counted under a DIC microscopy.

**Isolation of newborn cells by elutriation**. WT, *cip1*, *sic1*, and *cip1 sic1* strains were pre-grown in YEPD at 30 °C, diluted in YEP-raffinose and grown to OD$_{660}$ = 2 before elutriation. Elutriated cells were analyzed at the microscope, and newborn cells were pooled to reach OD$_{660}$ = 0.6 and recovered for 90–120 min in YEPD at 25 °C. At time 0, cultures were split into halves for 0.5 M KCl treatment. Cell cycle progression was analyzed by FACS analysis.

**Co-immunoprecipitation**. The co-immunoprecipitation assay was performed using yeast protease-deficient BJ2168 strains overexpressing Cip1 and cyclin in two different cells. Cells were grown to exponential phase in SC medium with 2% raffinose, and *GAL1* promoter-driven protein was induced by 2% galactose at 30 °C. 0.5 M KCl was added for 10 min at 30 °C to induce phosphorylation of Cip1. Cell pellets were lysed by lysis buffer (100 mM NaCl, 0.1% Triton X-100, 50 mM Tris-HCl pH 7.5). The solubilized fractions containing cyclin or Cip1 were mixed and incubated with the antibody-coupled Sepharose beads for 1 h. The binding proteins were separated and detected by western blotting.

**MS analysis**. To identify the phosphorylation sites of Cip1, GST-Cip1was overexpressed from pEGKT-*CIP1* in 0.5 M KCl treated yeast, isolated by pull-down and SDS–polyacrylamide gel electrophoresis (SDS–PAGE) followed by the in-gel enzyme digestion. The tryptic peptides of Cip1were analyzed by nanoscale liquid chromatography coupled to tandem mass spectrometry (nano LC-MS/MS) instrument (LTQ-FT, Thermo Fisher Scientific). The MS/MS spectra data were converted as mgf format from experiment RAW file by MM File Conversion Tools[67] (http://www.massmatrix.net) and then analyzed by MassMatrix[68] for MS/MS ion search. The search parameters in MassMatrix including the error tolerance of precursor ions and the MS/MS fragment ions in spectra were 10 p.p.m. and 0.6 Da. The enzyme for digestion was assigned to be trypsin with the miss cleavage number two. The variable post-translational modifications in search parameters were assigned to include the oxidation of methionine, carbamidomethylation of cysteine, and the phosphorylation of serine/threonine/tyrosine. The phosphorylation sites of Cip1 identified by MS were T65, T69, and T73 (Supplementary Fig. 5c).

**Phosphor-specific antibody preparation**. Phosphopeptides were conjugated with KLH protein and mixed with Freund's Incomplete Adjuvant (Thermo, Cat.77145) as antigen. New Zealand rabbit was boosted by subcutaneous injected with 0.5 mg per 2 ml antigen every 2 weeks until a sufficient titer. Three days before the final immunization, the rabbit was injected with 0.5 mg antigen without adjuvant into a marginal ear vein. The rabbit was then sacrificed, and blood was collected by cardiac puncture. The blood was incubated at 37 °C for 1 h to form clots and incubated at 4 °C overnight. To separate serum and blood clots, blood was centrifuged at 1,000×*g* for 30 min at 4 °C. The supernatant was collected and stored at −80 °C.

To improve the titer and specificity of the phosphor-specific antibody, phosphor- or non-phosphopeptides were conjugated with NHS-activated agarose (Thermo, Cat. 26196). After being purified and concentrated by phosphopeptides column and non-phosphopeptides column, the phosphor-specific antibody was stored in 50% glycerol, 0.05% sodium azide at −80 °C.

**Dot blot analysis**. A quantity of 50, 5, and 0.5 ng of the phosphorylated or unphosphorylated peptides were spotted on nitrocellulose membranes. Dot blot analysis was conducted by standard protocol using phosphor-specific antibodies.

**Western blot analysis**. Whole proteins were extracted and resolved by SDS–PAGE, and transferred to a polyvinylidene difluoride membrane. The membrane was blocked in 5% nonfat milk at room temperature for 1 h, followed by incubation with primary antibody at 4 °C overnight. Primary antibodies were used to detect HA (Roche, 12CA5, 1:1,000), Myc (Roche, 9E10, 1:1,000), β-actin (GeneTex, GTX109639, 1:1,000), Cip1 pT65 (1:500), and Cip1 pT73 (1:500). Signals were developed using Luminata™ Crescendo Western HRP Substrate (Millipore). The image was quantified by ImageJ software. All uncropped western blots can be found in Supplementary Fig. 11.

**In vitro kinase assay**. GST fusion proteins encoding for Cip1 wild-type, Cip1–3TA, Sic1, Hog1, and Pbs2[EE] were expressed in BL21(DE3) for 5 h at 16 °C. Cells were lysed by sonication in STET buffer (10 mM Tris pH 8.0, 100 mM NaCl, 1 mM EDTA pH 8.0, 5% Triton X-100, 2 mM dithiothreitol (DTT), 1 mM phenylmethanesulfonyl fluoride (PMSF), 1 mM benzamidine, 2 mg ml⁻¹ leupeptin, and 2 mg ml⁻¹ of pepstatin). Clear lysates were incubated with glutathione-Sepharose beads (GE Healthcare) and GST-tagged proteins were eluted in Elution buffer (50 mM Tris pH 9.5, 2 mM DTT, 10 mM reduced glutathione). One microgram of Hog1 was activated with 0.5 µg of Pbs2[EE] in the presence of kinase buffer (50 mM Tris-HCl pH 7.5, 10 mM MgCl$_2$, 2 mM DTT) and 100 µM ATP. After 30 min at 30 °C, 2 µg of Cip1 wild-type, Cip1–3TA or Sic1 was added to the Hog1/Pbs2[EE] mixture together with $^{32}$P-ATP (0.1 mCi per ml) and incubated for 30 min at 30 °C. Hog1 inhibitor SB203580 (Calbiochem) was added to the relevant samples to a final concentration of 10 µM prior to the addition of the substrates to the Hog1/Pbs2[EE] mixture. The reaction was terminated by the addition of Laemmli buffer and subsequent boiling. Proteins were resolved in 8% SDS–PAGE. Gels were stained with Coomassie blue, dried, and analyzed by autoradiography.

Cdk1 in vitro kinase assay was performed as described[20, 69]. Briefly, Cln1/Cdk1, Cln2/Cdk1, and Cln3/Cdk1 were immunoprecipitated from lysates prepared from BJ2168 bearing pGAL1-CLN1-3HA, pGAL1-CLN2-3HA, or pGAL1-CLN3-3HA, respectively. Recombinant GST and GST-Cip1 proteins were incubated with Cdk1/cyclin immunoprecipitates for 30 min at 30 °C, and histone H1 (Sigma, H4524) was used as a kinase substrate. Labeled proteins were separated by 12% SDS-PAGE, and $^{32}$P-labeled histone H1 was visualized by autoradiography and quantitated with a phosphoimager (Molecular Dynamics).

**Fluorescence microscopy**. Cip1–GFP is driven by CIP1 promoter from a pYEPFAT7 plasmid. Cells overexpressing Cip1–GFP were grown to exponential phase in SC-Ura minus medium. Cip1–GFP was induced in SC-Leu minus medium at 30 °C for 3 h. 0.5 M KCl was added and incubated at 30 °C for 5 min. Samples were centrifuged, fixed with 70% ice-cold ethanol, and washed twice with PBS. DAPI (4′,6-diamidino-2-phenylindole) was added to mark the nucleus. The cells were visualized on a Zeiss Imager.M2 fluorescence microscope.

**Data availability**. The data that supports the findings of this study are available within the article and its supplementary information files or from the corresponding author upon reasonable request.

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

## Acknowledgements

We thank Drs Jing-Jer Lin and Tsai-Kun Li for valuable inputs, and Drs George F. Sprague and Ji-Sook Hahn for providing materials. We also thank Mariona Nadal-Ribelles for helping in the protein purification and preparing materials of Hog1 *in vitro* kinase assay. This work was supported by grants from the Ministry of Science and Technology (MoST-105-2311-B-002-015-MY3), the National Taiwan University (NTU-ERP-106R8805A1) to Shu-Chun Teng, the Spanish Ministry of Economy and Competitiveness (BFU2015-64437-P and FEDER), the Catalan Government (2014 SGR 599), and the Fundación Botín, by Banco Santander through its Santander Universities Global Division to F.P. F.P. is recipients of an ICREA Acadèmia (Generalitat de Catalunya).

## Author contributions

Y.-L.C., S.-F.T., Y.-C.H., Z.-J.S., P.-H.H., M.-H.H., C.-W.Y., S.T., B.C., L.S., C.-W.W., H.-T.C., and C.-Y.L. conducted and analyzed the experiments. Y.-L.C., S.-F.T., Y.-C.H., Z.-J.S., and S.-C.T. designed the experiments and interpreted the data. Y.-L.C., Z.-J.S., F.P., and S.-C.T. edited the manuscript. S.-C.T. and F.P. provided conceptual advice. S.-C.T. conceived and supervised this study.
