## [Peer Review File · Nature Communications]

Reviewers' comments:

Reviewer #1 (Remarks to the Author):

The article by Chang et al. describes the role of Cip1, a recently identified Cdk inhibitor of budding yeast, in the G1 delay caused by osmotic stress. The authors first confirm that Cip1 inhibits Cdc28-Cln kinase activity in vitro while causing a G1 delay when overexpressed. Deleterious effects of CIP1 overexpression are SIC1 and FAR1 independent, but suppressed by co-expression of CLNs. Interestingly, CIP1 expression is strongly upregulated under osmotic and oxidative stress conditions involving Msn2/4 TFs and STRE sequences at the CIP1 promoter. In addition, Cip1 is phosphorylated by Hog1 in vitro and in vivo very rapidly after osmotic shock. A phosphoablated mutant, just as the *cip1* deletant, is slightly affected in the G1 delay caused by stress. Since the phosphoablated Cip1 protein binds less avidly to G1 cyclins, the authors propose a model whereby Hog1-mediated phosphorylation of Cip1 would increase its inhibitory role on Cdk-Cln complexes thus causing a delay in G1. While this work addresses an interesting question in cell biology, the main conclusions are not sufficiently supported by the experimental data. In my view, the authors should fully address the following issues before publication in Nat Commun might be considered.

Major points

1. Does CLN co-expression suppress the G1 delay caused by overexpressing CIP1? These experiments should be done by FACS analysis as in Fig. 1b.
2. The phosphoablated *cip1*-3TA mutant does not show any deleterious effects on growth when overexpressed under non-stress conditions. On the other hand, the G1 delay displayed by the *cip1*-3TA mutant under stress is very similar to that displayed by the *cip1* deletant. Both observations would fit with the possibility that the Cip1-3TA protein is in fact an inactive version of Cip1 intrinsically unable to bind G1 cyclins. This serious doubt could have been solved if the phosphomimetic mutant displayed the expected dominant behavior as Cdk inhibitor but, unfortunately, this seems not to be the case as its overexpression did not cause any growth defect.
3. The authors claim that "Cip1 phosphorylation promotes cell cycle arrest". However, the phosphomimetic mutant protein caused no effects on growth when overexpressed, just like the phosphoablated mutant. Thus, this important conclusion of the article is left without demonstration.
4. The G1 delay caused by osmotic stress is only slightly alleviated by a *cip1* deletion. The authors extended this genetic analysis to a *sic1* deletant, where they could observe a somewhat larger effect in budding efficiencies. However, since Sic1 is not involved in budding control, how do the authors interpret these results?
5. Does osmotic stress increase the interaction of wild-type Cip1 and G1 cyclins? This question stems from the main proposal of the article.
6. The effects of deleting CIP1 in the G1 delay (budding, Whi5 phosphorylation, cycle-trap assay) caused by osmotic stress are rather modest when analyzed in alpha-factor arrested cells. Since CIP1 is downregulated during the alpha-factor arrest, I would suggest testing its relevance in newborn cells (elutriated or obtained from Ficoll gradients) shocked at different time points after growth resumption in G1.

Minor points

7. The authors claim that CIP1 expression peaks at mid G1. Their data, as well as that published by Spellman et al. (1998), suggest a pattern very similar to CLN2, which is induced in late G1 (not mid G1). On the other hand, removal of ECBs does not abolish that pattern, suggesting that Mcm1 would ensure basal CIP1 expression levels in early G1 but not in late G1, where transcription would be further increased by MBF as the authors discuss.
8. How fast is CIP1 upregulated under osmotic stress (5, 10 min?)
9. The fact that Cdk-binding compromised Cln3 mutants bind Cip1 more efficiently does not support the idea of Cip1 being a "classical" CKI. This point should be discussed in the article.
10. Legends for Fig. 5e and 5f are swapped.

Reviewer #2 (Remarks to the Author):

This is an interesting work about Cip1 regulation under osmotic stress, after the identification of Cip1 as a CKI from other lab (Ren, et al. 2016). In this manuscript, the Teng lab examines that Cip1 interacts and inhibits all three G1 cyclins. The authors show that Cip1 is regulated by Mcm1 in unperturbed cell cycle and induced by Msn2/4 in response to osmotic stress. Moreover, they provide data to suggest that stress-activated protein kinase Hog1 directly phosphorylates Cip1 and thus inhibits Cln-CDK activity for early G1 progression in response to osmotic stress. Interestingly, they also draw a possibility that links yeast Cip1 to metazoans p21.

In general, it is a nice paper about the regulation and function study of the newly identified CKI Cip1. However, there are a few main issues with this paper as follows, which invalidates acceptance in Nature Communications in its current form.

-First, the data suggesting that Cip1 is directly targeted by Hog1 is not convincing.

1- Figure 5b showing Hog1 in vitro kinase assay is difficult to interpret for the conclusion. The recombinant Cip1 protein was loaded with too many co-purified contaminant proteins. Moreover, these co-purified bands even display stronger hot signals, although not with rcip1-3TA. In addition, inactive Hog1 control is lack. Plus, the dirty rCip1 would also disrupt the conclusion from Figure 1e, where rCip1 is used as the candidate inhibitor of Cln-CDK. However, the authors do not show full Coomassie Blue-stained gels (what if the co-purified contaminant proteins, as exist in Figure 5b experiments, block CDK activity). It is better to use more purified rCip1 for the in vitro experiments (Pure rGST-Cip1 is available in Ren, et al. 2016).

2- Moreover, they show phosphor-Cip1 western, and beta-actin but not total Cip1 protein in Figure 5c and 5d, which is not enough for the conclusion. Cip1 expression is regulated in both normal and osmotic stress conditions; therefore authors need to show the total Cip1 levels side by side as the essential control.

3- The authors claim that identified T65 and T73 phosphorylation is cell cycle independent. However, they do not show enough convincing data. Cip1 exists Clb-CDK dependent phosphorylation (Ren et al, 2016), but Hog1 and Cdk1 share the same consensus substrate motif (S/T-P), which requires authors to present more solid data to support that these two sites phosphorylation being Hog1 dependent but cell cycle independent. In supplementary Figure 4d and e, FACS control or Cln and Clb cyclin levels showing cell cycle phase is missing. And Authors should explain why Cip1 protein levels keep stable during the time course?

-Second, the author claims that Cip1 phosphorylation enhances its binding to G1-Cdk1. However, the data are only obtained mainly from experiments using artificial Cip1-3TA mutants.

4- Results from Figure 6a are not enough for the conclusion. Point mutant of amino acid would change the protein structure therefore affects the interaction, so it should not be presented as key data for the conclusion. It would be important to show the percentage of Co-IPed phosphorylated Cip1 in Cyclin IP experiments versus the percentage of input phosphorylated Cip1 from total Cip1. Authors have very specific phosphor-Cip1 antibodies; it is better and easy for them to do such experiments.

5-Importantly, the binding affinity of Cip1-CDK should be compared in a native condition in osmotic stress and in untreated condition, taking advantage of phosphor-Cip1 antibodies (instead of using artificial Cip1-3TA).

-Minor points

6- Since authors show that Cip1 is induced in response to multiple stresses, people would like to know whether *cip1-del* is sensitive to those stresses. I thus recommend that the authors describe or show data related to this concern. If it is not sensitive, then authors should discuss on this concern.

7- Authors should play around with the SDS-PAGE conditions to better separate Cip1-13myc, as in Ren et al paper, same Cip1-13myc fusion protein show clear cell cycle dependent phosphorylation shift. Authors have quite nice anti-phospho-Cip1 antibodies to compare the cell cycle dependent phosphorylation with Hog1 dependent phosphorylation in such conditions.

8- Authors draw a very interesting linker between yeast Cip1 and metazoans p21. If authors can present some more data to directly support such possibility, it would notably strengthen the quality of this work.

9- Some experimental procedures are not well described. For instance, what is the temperature condition for the time course in Supplementary Figure 4d and 4e?

10- Some supporting data are similar to the previous paper, authors should carefully describe them. For instance, Figure 1a, 1b and the case of *Cln2* in Figure 1e are the confirmation of work from Ren et al 2016.

11- Minor correction: BY4741 appears several times as the strain name for different strains in Supplementary Table 1 (*far1*, *sic1* *cip1*).

Reviewer #3 (Remarks to the Author):

Chang et al., "multiple pathways activate Cip1....."

The paper describes a comprehensive effort to elaborate on the role of the yeast Cip1 protein. Until recently this protein was known as a mere ORF in the yeast genome, but a recent study suggested that it functions as a CKI (cyclin dependent kinase inhibitor). As such, it is an important protein justifying the effort to understand its exact biochemical and cellular functions, its regulation and its mechanism of action.

The current study approaches these questions so that the paper could be divided into 4 parts.

In the first, the authors obtained support for the notion raised in a previous study that Cip1 is a CKI. The experimental approach relied heavily on cells overexpressing Cip1 under the GAL promoter and the outputs were monitoring rate of budding, release from G1 arrest and effect of co-overexpression of cyclins. These experiments support the previous work and cannot be considered as novel.

The second part dealt with the regulation of Cip1 transcription and the major finding were that its transcription is cell-cycle regulated so that it is highly expressed at mid G1 and its transcription is regulated by the transcription factor Mcm1. This was concluded following deletion of putative Mcm binding sites at the CIP promoter and ChIP experiment. This study is preliminary and under-developed. Required are detailed promoter analysis, EMSA with purified Mcm, effect of overexpression of Mcm on CIP1 and more...

The third part addressed the expression and function of Cip1 under stresses, primarily under high osmotic pressure. This part is more developed and the findings are more interesting. It was found that Cip1 is regulated by stress-responsive MAP kinase Hog1 and serves as the direct substrate of this MAPK. The phosphoacceptors of Hog1 on Cip1 were mapped via mass-spec analysis and relevant antibodies were raised. This part also showed that transcription of CIP under stress is dependent on the transcription factors Msn2 and Msn4.

The fourth part looked into the possible mechanism through which Cip1 controls cell cycle and

describes its interaction with Cln3 and inhibiting the activity of Cdk1-Cln3 complex.

The strength of the work is that it provides a comprehensive information on a seemingly important protein, on which little is known. This is also the weakness of the work. Since it tried to address 4 different questions, most of the answers are partial and uncompleted. The third part, which deals with the function of Cip1 under osmostress and its relationships with Hog1 is more developed than others. I feel that focusing on this part would yield a deeper, more conclusive, convincing and coherent study that may merit publication in a leading journal. But a serious work is needed to reach this point.

More specific comments

The fact that FACS analysis does not show any difference between wt and *cipΔ*, in other words FACS results do not coincide with the budding index data reduces excitement with respect to the concept that Cip is controlling cell cycle under osmostress. This is critical, the authors must give a deeper thought here to explain these results. Further experiments are clearly needed here, which may change the model...)

The result with 4E and 4A is confusing - the phospho-mimetic mutations and the mutations that make Cip un-phosphorylatable have the same effect on Cip1 ability to suppress growth (Fig....). A good explanation should be provided here...

If Hog1-mediated Cip phosphorylation is important for adaptation to high osmotic pressure what are the phenotypes of *cipΔ* under some osmostressors? What is Cip localization under osmostress? Does this localization change in *hog1Δ*? What are its relationships with Sic1 - more assays are necessary with the double knockout..

The finding that Msn2 and Msn4 are necessary for CIP1 transcription under osmotic pressure is convincing. This result calls for looking at the relevant upstream signaling cascades. Ras/adenylyl cyclase, Gpa1/adenylyl, Tor and Pkc pathways are known suppressors of Msn2 and 4. Which is relevant here? The authors can look at Cip1 expression and activity in some of the related mutants. Would Cip be spontaneously transcribed and active, in *rasΔ* strains? In mutants of the Tor or PKC cascades? How would this mutant behave in response to osmotic pressure. How would they behave if CIP is eliminated from them??

Although the effect of eliminating Msn2 and 4 is convincing and CIP expression in cells lacking those factors is indeed low, only a minimal effort was done to study mechanisms. Reporter system under the CIP1 promoter should be constructed and used here, EMSA and more promoter analyses...

The final model claims that in the absence of stress Cip1 transcription is regulated by Mcm1 whereas under stress by Msn2 and Msn4. If so, Mcm1 is not relevant under stress and Msn2 and 4 are not relevant under other conditions (?). This should be shown.. The authors actually established some reagents to test this idea (deletion promoters, knockout strains, ChIP protocols) why didn't they test it???

Point-to-point Response to Reviewers' Comments

- For the ease of reading, original remarks from the reviewers are marked in blue, and responses by the authors are marked in black.

Reviewers' comments:

Reviewer #1 (Remarks to the Author):

The article by Chang et al. describes the role of Cip1, a recently identified Cdk inhibitor of budding yeast, in the G1 delay caused by osmotic stress. The authors first confirm that Cip1 inhibits Cdc28-Cln kinase activity in vitro while causing a G1 delay when overexpressed. Deleterious effects of CIP1 overexpression are SIC1 and FAR1 independent, but suppressed by co-expression of CLNs. Interestingly, CIP1 expression is strongly upregulated under osmotic and oxidative stress conditions involving Msn2/4 TFs and STRE sequences at the CIP1 promoter. In addition, Cip1 is phosphorylated by Hog1 in vitro and in vivo very rapidly after osmotic shock. A phosphoablated mutant, just as the *cip1* deletant, is slightly affected in the G1 delay caused by stress. Since the phosphoablated Cip1 protein binds less avidly to G1 cyclins, the authors propose a model whereby Hog1-mediated phosphorylation of Cip1 would increase its inhibitory role on Cdk-Cln complexes thus causing a delay in G1.

While this work addresses an interesting question in cell biology, the main conclusions are not sufficiently supported by the experimental data. In my view, the authors should fully address the following issues before publication in Nat Commun might be considered.

Major points

1. Does CLN co-expression suppress the G1 delay caused by overexpressing CIP1? These experiments should be done by FACS analysis as in Fig. 1b.

Response 1. We have used FACS analysis to investigate the DNA content of cells with *CIP1* and *CLNs* co-expression. The results of G1 delay and recovery have been confirmed and incorporated in Fig. 1d and the Results (line 118-120).

2. The phosphoablated *cip1-3TA* mutant does not show any deleterious effects on growth when overexpressed under non-stress conditions. On the other hand, the G1 delay displayed by the *cip1-3TA* mutant under stress is very similar to that displayed

by the *cip1* deletant. Both observations would fit with the possibility that the Cip1-3TA protein is in fact an inactive version of Cip1 intrinsically unable to bind G1 cyclins. This serious doubt could have been solved if the phosphomimetic mutant displayed the expected dominant behavior as Cdk inhibitor but, unfortunately, this seems not to be the case as its overexpression did not cause any growth defect.

Response 2. It is not unusual that phosphomimetic mutants do not adequately mimic the structure of phospho-threonine. The negative charge introduced by glutamate substitutions (-1) does not match with that of the phosphorylated residue (-2) at physiological pH (Strickfaden et al. 2007). We have incorporated this explanation in the Discussion (line 414-419). Moreover, in the new Fig. 7a, we also showed that osmotic pressure increases the Cip1-Cln interaction, implying the importance of the osmotic stress-induced Cip1 phosphorylation on the Cip1-Cln interaction.

3. The authors claim that "Cip1 phosphorylation promotes cell cycle arrest". However, the phosphomimetic mutant protein caused no effects on growth when overexpressed, just like the phosphoablated mutant. Thus, this important conclusion of the article is left without demonstration.

Response 3. As mentioned above in Response 2, also from our previous experience, phosphomimetic mutation does not always reflect the original structure of the phosphorylated residue. A possible explanation has been discussed. Additional evidence of osmotic stress-stimulated Cip1-Cln interaction has been included in Fig. 7a to support our model.

4. The G1 delay caused by osmotic stress is only slightly alleviated by a *cip1* deletion. The authors extended this genetic analysis to a *sic1* deletant, where they could observe a somewhat larger effect in budding efficiencies. However, since Sic1 is not involved in budding control, how do the authors interpret these results?

Response 4. The budding index represents the fraction of budded cells in an exponentially growing yeast culture. Here we use budding index to monitor the percentage of cells that exits the G1 (unbudded) phase in the indicated strains. Deletion of *SIC1* accelerates S phase entry under osmotic stress, thereby increasing the percentage of budded cells. We have added this description in the Results (line 282) to clarify this misunderstanding.

5. Does osmotic stress increase the interaction of wild-type Cip1 and G1 cyclins? This

question stems from the main proposal of the article.

Response 5. This is a wonderful suggestion, which was also raised by Reviewer #2. We have now performed additional experiments and assessed the interaction of wild-type Cip1 and G1 cyclins under osmotic stress. In cells that were subjected to osmotic stress, Cip1 exhibited higher binding affinity to each of the G1 cyclins than that in untreated cells. These findings have been included in Fig. 7a and in the Results (line 318-319).

6. The effects of deleting CIP1 in the G1 delay (budding, Whi5 phosphorylation, cycle-trap assay) caused by osmotic stress are rather modest when analyzed in alpha-factor arrested cells. Since CIP1 is downregulated during the alpha-factor arrest, I would suggest testing its relevance in newborn cells (elutriated or obtained from Ficoll gradients) shocked at different time points after growth resumption in G1.

Response 6. As suggested by the reviewer, we elutriated wild-type, *cip1*, *sic1* and *cip1 sic1* cells, and the newborn cells were selected to analyze cell cycle progression under osmotic stress. The results support our initial observations and they have now been included in the manuscript (Fig. 6b) (line 283-289).

Minor points

7. The authors claim that CIP1 expression peaks at mid G1. Their data, as well as that published by Spellman et al. (1998), suggest a pattern very similar to CLN2, which is induced in late G1 (not mid G1). On the other hand, removal of ECBs does not abolish that pattern, suggesting that Mcm1 would ensure basal CIP1 expression levels in early G1 but not in late G1, where transcription would be further increased by MBF as the authors discuss.

Response 7. We have changed the description from “with peak expression at mid G1” to “with peak expression at G1” to improve the clarity.

8. How fast is CIP1 upregulated under osmotic stress (5, 10 min?)

Response 8. The *CIP1* expression under osmotic stress was monitored in more detail by qRT-PCR. Indeed, its level starts to elevate even at 5 minutes after KCl treatment. The results have been incorporated into Supplementary Fig. 3f and the Results (line 163-164).

9. The fact that Cdk-binding compromised Cln3 mutants bind Cip1 more efficiently does not support the idea of Cip1 being a "classical" CKI. This point should be discussed in the article.

Response 9. Based on the structure and CDK targets, CKIs are divided into two families. The canonical Kip family CKIs bind both the cyclin and the CDK to prevent the CDK-cyclin complexes kinase activity. In contrast, the INK4 family proteins specifically bind CDK monomers, which twist the CDK to impede cyclin binding and kinase activity. Overexpressed G1 cyclins in a cell can partially overcome the Cip1 blockage of these cells in G1. On the other hand, Cip1 displays greater association with free form Cln3 that the Cip1 binding on Cln3 may block Cdk1 to form a complex with Cln3. Taken together, we do not consider Cip1 as a canonical CKI. We have added this description in the Discussion as suggested by the reviewer (line 422-429).

10. Legends for Fig. 5e and 5f are swapped.

Response 10. The description in Figure Legends has been corrected in the manuscript as suggested by the reviewer (line 1020-1022 and line 1030-1034).

Reviewer #2 (Remarks to the Author):

This is an interesting work about Cip1 regulation under osmotic stress, after the identification of Cip1 as a CKI from other lab (Ren, et al. 2016). In this manuscript, the Teng lab examines that Cip1 interacts and inhibits all three G1 cyclins. The authors show that Cip1 is regulated by Mcm1 in unperturbed cell cycle and induced by Msn2/4 in response to osmotic stress. Moreover, they provide data to suggest that stress-activated protein kinase Hog1 directly phosphorylates Cip1 and thus inhibits Cln-CDK activity for early G1 progression in response to osmotic stress. Interestingly, they also draw a possibility that links yeast Cip1 to metazoans p21.

In general, it is a nice paper about the regulation and function study of the newly identified CKI Cip1. However, there are a few main issues with this paper as follows, which invalidates acceptance in Nature Communications in its current form.

-First, the data suggesting that Cip1 is directly targeted by Hog1 is not convincing.

1- Figure 5b showing Hog1 in vitro kinase assay is difficult to interpret for the conclusion. The recombinant Cip1 protein was loaded with too many co-purified contaminant proteins. Moreover, these co-purified bands even display stronger hot signals, although not with rcip1-3TA. In addition, inactive Hog1 control is lack. Plus, the dirty rCip1 would also disrupt the conclusion from Figure 1e, where rCip1 is used as the candidate inhibitor of Cln-CDK. However, the authors do not show full Coomassie Blue-stained gels (what if the co-purified contaminant proteins, as exist in Figure 5b experiments, block CDK activity). It is better to use more purified rCip1 for the in vitro experiments (Pure rGST-Cip1 is available in Ren, et al. 2016).

Response 1. Following the reviewer suggestion we have now purified cleaner recombinant Cip1 for the in vitro Hog1 kinase assay and also included a Hog1 inhibitor (SB203580) as a negative control for Hog1 activity. The overall assay has also been optimized and bigger size gels have been used to resolve the kinase assays better. The new results support our initial observations (line 235-240) and the new figure has been included as Figure 5b. We have also included the full-length Coomassie blue-stained images for the Fig. 1e in vitro Cdk1 kinase assay in Supplementary Fig. 1, in which reaction also contains purified recombinant Cip1.

2- Moreover, they show phosphor-Cip1 western, and beta-actin but not total Cip1 protein in Figure 5c and 5d, which is not enough for the conclusion. Cip1 expression

is regulated in both normal and osmotic stress conditions; therefore authors need to show the total Cip1 levels side by side as the essential control.

Response 2. We have included total Cip1 levels as the loading control as suggested by the reviewer. The result has been incorporated into Fig. 5c.

3- (a) The authors claim that identified T65 and T73 phosphorylation is cell cycle independent. However, they do not show enough convincing data. Cip1 exists Clb-CDK dependent phosphorylation (Ren et al, 2016), but Hog1 and Cdk1 share the same consensus substrate motif (S/T-P), which requires authors to present more solid data to support that these two sites phosphorylation being Hog1 dependent but cell cycle independent.

Response 3a. We have determined the phosphorylation of Cip1 T65 and T73 in the *cdc28-as1* strain treated with or without the Cdk1 inhibitor, 1-NM PP1, under osmotic shock. The inhibition of Cdk1 activity displayed no influence on Cip1 T65 and T73 phosphorylation under osmotic stress treatment. The results have been incorporated into Supplementary Fig. 7e and the Results (line 251-257).

(b) In supplementary Figure 4d and e, FACS control or Cln and Clb cyclin levels showing cell cycle phase is missing.

Response 3b. We have included FACS analysis as the cell-cycle phase control as suggested by the reviewer. The result has been incorporated into Supplementary Fig. 7d (Supplementary Fig. 4d and 4e in the original manuscript).

(c) And Authors should explain why Cip1 protein levels keep stable during the time course?

Response 3c. The Supplementary Fig. 7d (Supplementary Fig. 4d and 4e in the original manuscript) has been replaced by a clearer gel in which Cip1 expression level peaks at G1 and gradually declines after cells enter S phase.

-Second, the author claims that Cip1 phosphorylation enhances its binding to G1-Cdk1. However, the data are only obtained mainly from experiments using artificial Cip1-3TA mutants.

Response 3d. We have determined the Cip1-Cln interaction under osmotic stress by

co-immunoprecipitation. Upon osmotic stress, cyclins pull-down more wild-type Cip1, indicating that stress-induced Cip1 phosphorylation strengthens the Cip1-Cln interaction. The new data have been incorporated in Fig. 7a.

4- Results from Figure 6a are not enough for the conclusion. Point mutant of amino acid would change the protein structure therefore affects the interaction, so it should not be presented as key data for the conclusion. It would be important to show the percentage of Co-IPed phosphorylated Cip1 in Cyclin IP experiments versus the percentage of input phosphorylated Cip1 from total Cip1. Authors have very specific phosphor-Cip1 antibodies; it is better and easy for them to do such experiments.

Response 4. Our phosphor-specific antibodies can detect Cip1 phosphorylation from TCA-treated lysates. TCA buffer denatures all phosphatases and stabilizes phosphorylation. We have tried to determine the phosphorylated Cip1 from co-immunoprecipitated Cip1 by our Cip1 phosphor-specific antibodies many times. Unfortunately, we could only detect faint phosphor-signal in the input samples, but not in the co-immunoprecipitated samples. Due to that the amount of phosphorylated Cip1 in inputs is about one tenth to that that in TCA samples, and the efficiency of immunoprecipitation is usually less than 5%, the phosphorylated Cip1 in co-immunoprecipitated samples was below the limit of detection. This limitation of our Cip1 phosphor-specific antibodies made us unsuccessfully detecting phosphorylated Cip1 in co-immunoprecipitated samples.

5-Importantly, the binding affinity of Cip1-CDK should be compared in a native condition in osmotic stress and in untreated condition, taking advantage of phosphor-Cip1 antibodies (instead of using artificial Cip1-3TA).

Response 5. This is an excellent suggestion which was also raised by Reviewer #1. The interaction of wild-type Cip1 and G1 cyclins has been compared in untreated and osmotic stress condition. When cells are subjected to osmotic stress, Cip1 exhibited higher binding affinity to each G1 cyclin than in untreated cells. These findings have been included in Fig. 7a and in the Results (line 318-319). From our results in Fig. 5c, phosphorylated Cip1 dramatically increases in cells under osmotic shock, but not in untreated cells. Altogether, we postulate that the increase of phosphorylated Cip1 under osmotic stress augmented co-immunoprecipitated Cip1.

-Minor points

6- Since authors show that Cip1 is induced in response to multiple stresses, people would like to know whether *cip1-del* is sensitive to those stresses. I thus recommend that the authors describe or show data related to this concern. If it is not sensitive, then authors should discuss on this concern.

Response 6. We have determined the stress sensitivity of *CIP1* deleted cells as suggested by the reviewer. *cip1* cells were not sensitive to these stresses. The results have been incorporated into Fig. 6c, Supplementary Fig. 4a,4b and the Results (line 164-166 and line 289-292), and a possible explanation, such as the redundancy with Sic1, has also been included in the Discussion (line 400-407).

7- Authors should play around with the SDS-PAGE conditions to better separate Cip1-13myc, as in Ren et al paper, same Cip1-13myc fusion protein show clear cell cycle dependent phosphorylation shift. Authors have quite nice anti-phospho-Cip1 antibodies to compare the cell cycle dependent phosphorylation with Hog1 dependent phosphorylation in such conditions.

Response 7. We have investigated the phosphorylation shift of Cip1-Myc₁₃ fusion protein by using different percentages of SDS-PAGE gel as suggested by the reviewer. Thanks to the reviewer's suggestion. We have now obtained better results that have been included in Fig. 5c and Supplementary Fig. 7d.

8- Authors draw a very interesting linker between yeast Cip1 and metazoans p21. If authors can present some more data to directly support such possibility, it would notably strengthen the quality of this work.

Response 8. More data has been included to support the link between yeast Cip1 and metazoans p21 as suggested by the reviewer. The spotting assay indicated that overexpression of human p21 in yeast resulted in inhibition of cell growth. This growth retardation was caused by G1 cell cycle arrest as determined by FACS analysis. The results have been included in Supplementary Fig. 10 and the Results (line 366-374).

9- Some experimental procedures are not well described. For instance, what is the temperature condition for the time course in Supplementary Figure 4d and 4e?

Response 9. We have added the temperature condition of the time course in Figure Legends of Supplementary Figure 7d (Supplementary Fig. 4d and 4e in the original

manuscript). Also, many experimental procedures have been strengthened as suggested by the reviewer.

10- Some supporting data are similar to the previous paper, authors should carefully describe them. For instance, Figure 1a, 1b and the case of Cln2 in Figure 1e are the confirmation of work from Ren et al 2016.

Response 10. We have added the related reference and strengthened the difference between our discovery and the findings in the previous paper in the Results (line 101, 104-105, and 122-123).

11- Minor correction: BY4741 appears several times as the strain name for different strains in Supplementary Table 1 (far1, sic1 cip1).

Response 11. The strain name has been corrected as suggested by the reviewer (Supplementary Table 1).

Reviewer #3 (Remarks to the Author):

Chang et al., "multiple pathways activate Cip1....."

The paper describes a comprehensive effort to elaborate on the role of the yeast Cip1 protein. Until recently this protein was known as a mere ORF in the yeast genome, but a recent study suggested that it functions as a CKI (cyclin dependent kinase inhibitor). As such, it is an important protein justifying the effort to understand its exact biochemical and cellular functions, its regulation and its mechanism of action. The current study approaches these questions so that the paper could be divided into 4 parts.

In the first, the authors obtained support for the notion raised in a previous study that Cip1 is a CKI. The experimental approach relied heavily on cells overexpressing Cip1 under the GAL promoter and the outputs were monitoring rate of budding, release from G1 arrest and effect of co-overexpression of cyclins. These experiments support the previous work and cannot be considered as novel.

The second part dealt with the regulation of Cip1 transcription and the major finding were that its transcription is cell-cycle regulated so that it is highly expressed at mid G1 and its transcription is regulated by the transcription factor Mcm1. This was concluded following deletion of putative Mcm binding sites at the CIP promoter and ChIP experiment. This study is preliminary and under-developed. Required are detailed promoter analysis, EMSA with purified Mcm, effect of overexpression of Mcm on CIP1 and more...

The third part addressed the expression and function of Cip1 under stresses, primarily under high osmotic pressure. This part is more developed and the findings are more interesting. It was found that Cip1 is regulated by stress-responsive MAP kinase Hog1 and serves as the direct substrate of this MAPK. The phosphoacceptors of Hog1 on Cip1 were mapped via mass-spec analysis and relevant antibodies were raised. This part also showed that transcription of CIP under stress is dependent on the transcription factors Msn2 and Msn4.

The fourth part looked into the possible mechanism through which Cip1 controls cell cycle and describes its interaction with Cln3 and inhibiting the activity of Cdk1-Cln3 complex.

The strength of the work is that it provides a comprehensive information on a seemingly important protein, on which little is known. This is also the weakness of the work. Since it tried to address 4 different questions, most of the answers are partial and uncompleted. The third part, which deals with the function of Cip1 under osmotic stress and its relationships with Hog1 is more developed than others. I feel that focusing on this part would yield a deeper, more conclusive, convincing and coherent

study that may merit publication in a leading journal. But a serious work is needed to reach this point.

More specific comments

1. The fact that FACS analysis does not show any difference between wt and *cipΔ*, in other words FACS results do not coincide with the budding index data reduces excitement with respect to the concept that Cip is controlling cell cycle under osmostress. This is critical, the authors must give a deeper thought here to explain these results. Further experiments are clearly needed here, which may change the model...)

Response 1. We have determined cell cycle progression through analyzing the DNA content using FACS analysis in *sic1* and *cip1 sic1* cells. Loss of *CIP1* in *sic1* cells further accelerated cell cycle progression after osmotic stress treatment. The results reveal that Cip1 collaborates with Sic1 in osmotic stress-induced G1 cell cycle arrest. The FACS analysis shows no difference between wild-type and *cip1*, probably because the redundant CKI, Sic1 still induces a delay in cell cycle progression that overrides the effect of *cip1*. The results have been included in Fig. 6b and 6c and the Results (line 283-294).

2. The result with 4E and 4A is confusing - the phospho-mimetic mutations and the mutations that make Cip un-phosphorylatable have the same effect on Cip1 ability to suppress growth (Fig....). A good explanation should be provided here...

Response 2. It is not unusual that phosphomimetic mutants do not entirely mimic the structure of phosphor-threonine. The negative charge introduced by glutamate substitutions (-1) does not match with that of the phosphorylated residue (-2) at physiological pH (Strickfaden et al. 2007). It may be the reason why phosphomimetic mutations of Cip1 failed to reproduce the inhibitory phenotype caused by phosphorylation. We have incorporated this explanation in the Discussion (line 414-419).

3. (a) If Hog1-mediated Cip phosphorylation is important for adaptation to high osmotic pressure what are the phenotypes of *cipΔ* under some osmostressors?

Response 3a. We have determined the sensitivity of *CIP1* deleted cells under osmotic stress as suggested by the reviewer. *cip1* cells were not sensitive to osmotic pressure. However, as mentioned before, it made *sic1* cells much more sensitive to osmostress

indicating a role for Cip1 and a redundancy between those two factors. The results have been incorporated into Fig. 6c, Supplementary Fig. 4a, and the Results (line 164-166 and line 289-292), and has also been discussed in the Discussion (line 400-407).

(b) What is Cip localization under osmostress? Does this localization change in *hog1Δ*?

Response 3b. The localization of Cip1 under osmotic stress has been determined by fluorescence microscope examination. Cip1 localizes to the nucleus after osmotic stress, and this nuclear localization is Hog1-independent. The results have been incorporated in Supplementary Fig. 9 and the Results (line 354-365).

(d) What are its relationships with Sic1 - more assays are necessary with the double knockout.

Response 3d. The relationship between Cip1 and Sic1 has been further determined in stress sensitivity and FACS analysis by using *cip1 sic1* double knockout cells. The results have been included into Fig. 6b, Fig. 6c, Supplementary Fig. 4b, and the Results (line 284-294), and have also been discussed in the Discussion (line 430-444).

4. The finding that Msn2 and Msn4 are necessary for CIP1 transcription under osmotic pressure is convincing. This result calls for looking at the relevant upstream signaling cascades. Ras/adenylyl cyclase, Gpa1/adenylyl, Tor and Pkc pathways are known suppressors of Msn2 and 4. Which is relevant here? The authors can look at Cip1 expression and activity in some of the related mutants. Would Cip be spontaneously transcribed and active, in *rasΔ* strains? In mutants of the Tor or PKC cascades? How would this mutant behave in response to osmotic pressure. How would they behave if CIP is eliminated from them??

Response 4. The *CIP1* expression in RAS, GPA and TOR mutated cells and the response of these mutants to osmotic stress have been determined as suggested by the reviewer. *CIP1* expression was slightly increased in RAS, GPA or TOR pathway-abolished cells and was apparently induced in *gpa1* and *gpa2* cells. Neither these deletions nor *CIP1* elimination from them had an influence on cell growth under treatment with or without osmotic stress. The results have been incorporated into Supplementary Fig. 5a, 5b, and the Results (line 180-188). Since multiple pathways control Msn2/Msn4 repression, we feel that single blockade of one of these pathways

may not cause drastic influence. Since we could not find any reference demonstrating PKC pathway as an upstream suppressor of Cip1, we did not include PKC mutant cells in our experiment.

5. Although the effect of eliminating Msn2 and 4 is convincing and CIP expression in cells lacking those factors is indeed low, only a minimal effort was done to study mechanisms. Reporter system under the CIP1 promoter should be constructed and used here, EMSA and more promoter analyses...

Response 5. The *CIP1* promoter-driven Reporter system and EMSA have been performed as suggested by the reviewer. The results have been incorporated into Fig. 4b and 4c and the Results (line 191-197).

6. The final model claims that in the absence of stress Cip1 transcription is regulated by Mcm1 whereas under stress by Msn2 and Msn4. If so, Mcm1 is not relevant under stress and Msn2 and 4 are not relevant under other conditions (?). This should be shown. The authors actually established some reagents to test this idea (deletion promoters, knockout strains, ChIP protocols) why didn't they test it???

Response 6. The ChIP assay was conducted to determine the binding of Msn2 under oxidative stress and carbon source starvation, as well as to determine the binding of Mcm1 under osmotic stress. The data revealed that, under oxidative stress and carbon source starvation, Msn2 increased the binding affinity to the *CIP1* promoter. On the other hand, under osmotic pressure, the binding of Mcm1 at the *CIP1* promoter was reduced. These results have been incorporated into Fig. 2e and 3f-h and the Results (line 175-177).

References

Strickfaden, S. C., M. J. Winters, G. Ben-Ari, R. E. Lamson, M. Tyers, and P. M. Pryciak. 2007. A mechanism for cell-cycle regulation of MAP kinase signaling in a yeast differentiation pathway, *Cell*, 128: 519-31.

Reviewers' comments:

Reviewer #1 (Remarks to the Author):

The authors have made a significant effort and appropriately answered most of my questions and concerns. In particular, they have carried out key experiments in elutriated cells with clear results supporting the notion that Cip1 contributes with Sic1 to arrest cells in G1. Moreover, they have demonstrated a stronger interaction between Cdk-Clns and Cip1 under stress conditions, which greatly strengthens the main conclusion of the paper. As a minor point, and in view of the partial suppression of Cip1 overexpression effects by Cln overexpression, I would strongly suggest the authors to discuss the likely possibility that Cip1 were also a Cdk-Clb inhibitor. That would perfectly match the additive role of Cip1 and Sic1 in most functional cell assays, and considering the strong induction by MMS, it would open new possibilities for Cip1 as a key factor to prevent S phase entry under genotoxic conditions.

Reviewer #2 (Remarks to the Author):

The new version of manuscript from the Teng lab has some significant improvements. I thus would like to support publication of the paper. However, it is much better to see a direct evidence to one conclusion "Hog1-mediated Cip1 phosphorylation enhances the association between Cip1 and Cdk1-G1 cyclin complexes". This main conclusion is the key for understanding the mechanism of Cip1 function as an inhibitor of Cdk1 in response to osmotic stress.

Although authors tried many times to this point, their Co-IP experiments with anti-phosphor-Cip1 antibodies were not successful due to the limitation of the antibodies. Since the authors have already got very nice phosphorylation shift of Cip1-Myc13 on the SDS-PAGE gel, they could easily analyze phosphorylation shift of co-IPed Cip1 with G1 cyclin, instead of using anti-phosphor-Cip1 antibodies. I encourage authors to give a try.

Point-to-point Response to Reviewers' Comments

- For the ease of reading, original remarks from the reviewers are marked in blue, and responses by the authors are marked in black.

Reviewers' comments:

Reviewer #1 (Remarks to the Author):

The authors have made a significant effort and appropriately answered most of my questions and concerns. In particular, they have carried out key experiments in elutriated cells with clear results supporting the notion that Cip1 contributes with Sic1 to arrest cells in G1. Moreover, they have demonstrated a stronger interaction between Cdk-Clns and Cip1 under stress conditions, which greatly strengthens the main conclusion of the paper. As a minor point, and in view of the partial suppression of Cip1 overexpression effects by Cln overexpression, I would strongly suggest the authors to discuss the likely possibility that Cip1 were also a Cdk-Clb inhibitor. That would perfectly match the additive role of Cip1 and Sic1 in most functional cell assays, and considering the strong induction by MMS, it would open new possibilities for Cip1 as a key factor to prevent S phase entry under genotoxic conditions.

Response: We have incorporated the possibility that Cip1 may also be a Cdk-Clb inhibitor to prevent S phase entry under genotoxic conditions in the Discussion as suggested by the reviewer (line 435-437).

Reviewer #2 (Remarks to the Author):

The new version of manuscript from the Teng lab has some significant improvements. I thus would like to support publication of the paper. However, it is much better to see a direct evidence to one conclusion “Hog1-mediated Cip1 phosphorylation enhances the association between Cip1 and Cdk1-G1 cyclin complexes”. This main conclusion is the key for understanding the mechanism of Cip1 function as an inhibitor of Cdk1 in response to osmotic stress. Although authors tried many times to this point, their Co-IP experiments with anti-phosphor-Cip1 antibodies were not successful due to the limitation of the antibodies. Since the authors have already got very nice phosphorylation shift of Cip1-Myc₁₃ on the SDS-PAGE gel, they could easily analyze phosphorylation shift of co-IPed Cip1 with G1 cyclin, instead of using anti-phosphor-Cip1 antibodies. I encourage authors to give a try.

Response: As suggested by the reviewer, we have attempted to detect co-immunoprecipitated Cip1-Myc₁₃ by an antibody against Myc. However, the phosphorylation shift of Cip1 could only be detected from the TCA-treated total lysates in our overexpression system for co-IP (Response Letter Figure 1), probably because the TCA buffer denatures all phosphatases and thereby stabilizes the phosphorylation. Even though overproduction of Cip1 in cells might cause some stress that resulted in Cip1 hypophosphorylation, additional osmotic treatment further stimulated Cip1 phosphorylation. We tried all means to optimize our co-immunoprecipitation procedure and lysis buffer recipe with an excess of phosphatase inhibitors and wished to determine the phosphorylation shift of Cip1 from co-immunoprecipitated Cip1 numerous times. Unfortunately, the phosphorylation and phosphorylation shift of Cip1 are significantly decreased in the input samples immediately after cell breakage (Response Letter Figure 1). It is worthy to note that we have found many other osmotic stress-independent phosphorylation residues of Cip1. The difficulty to keep a large amount of phosphorylated Cip1 made us unsuccessfully to detect the phosphorylation shift of Cip1 in the Cln-HA co-immunoprecipitated samples by the anti-Myc antibody detected mobility shift.

Response Letter Figure 1. The phosphorylation shift of Cip1 is significantly decreased in the input samples right after cell breakage.

Cells overexpressing Cip1-Myc₁₃ were treated with or without 0.5 M KCl. The phosphorylation of Cip1 was detected by Western analysis using phosphor-specific antibodies against pT65 and pT73, or through gel mobility shift using an antibody against Myc in TCA-treated (lane 1 and 2), input (samples right after cell breakage, lane 3 and 4), and supernatant (samples after the co-immunoprecipitated procedure, lane 5 and 6) lysates. The phosphorylated Cip1 was indicated by asterisks (*, hypophosphorylated Cip1; **, hyperphosphorylated Cip1). An equal amount of lysates was loaded in each lane. The molecular weights are marked at the right. Shorter and longer exposed images are shown at the top and bottom, respectively.

Reviewer #3 (Remarks from Reviewer #1 to the Author):

Reviewer #3 was not able to send a report this time. We have asked Reviewer #1 to inform us whether the concerns from Reviewer #3 had been addressed and in confidential comments to the editor this reviewer states that most of the concerns have been addressed, but while the microscopy images show that Cip1 accumulates in the nucleus under osmotic stress, images do not clearly show a full independence of Hog1.

Response: Although in *hog1* cells Cip1 was still able to accumulate in the nucleus under osmotic stress, we have softened our conclusion and removed the description of “this nuclear localization is Hog1-independent” from the Results (line 357-358).